# Insects, Rodents, and Pets as Reservoirs, Vectors, and Sentinels of Antimicrobial Resistance

**DOI:** 10.3390/antibiotics10010068

**Published:** 2021-01-12

**Authors:** Willis Gwenzi, Nhamo Chaukura, Norah Muisa-Zikali, Charles Teta, Tendai Musvuugwa, Piotr Rzymski, Akebe Luther King Abia

**Affiliations:** 1Biosystems and Environmental Engineering Research Group, Department of Agricultural and Biosystems Engineering, University of Zimbabwe, Mount. Pleasant, Harare P.O. Box MP167, Zimbabwe; 2Department of Physical and Earth Sciences, Sol Plaatje University, Kimberley 8300, South Africa; nhamo.chaukura@spu.ac.za; 3Department of Environmental Sciences and Technology, School of Agricultural Sciences and Technology, Chinhoyi University of Technology, Private Bag, Chinhoyi 7724, Zimbabwe; norahmuisa@gmail.com or; 4Future Water Institute, Faculty of Engineering & Built Environment, University of Cape Town, Cape Town 7700, South Africa; charles.teta@uct.ac.za; 5Department of Biological and Agricultural Sciences, Sol Plaatje University, Kimberley 8300, South Africa; tendai.musvuugwa@spu.ac.za; 6Department of Environmental Medicine, Poznan University of Medical Sciences, 60-806 Poznan, Poland; rzymskipiotr@ump.edu.pl; 7Integrated Science Association (ISA), Universal Scientific Education and Research Network (USERN), 60-806 Poznań, Poland; 8Antimicrobial Research Unit, College of Health Sciences, University of KwaZulu-Natal, Private Bag X54001, Durban 4000, South Africa

**Keywords:** antimicrobial-resistant microorganisms, antimicrobial resistance genes, companion animals, human exposure pathways, human health risks, quantitative microbial risk assessment

## Abstract

This paper reviews the occurrence of antimicrobial resistance (AMR) in insects, rodents, and pets. Insects (e.g., houseflies, cockroaches), rodents (rats, mice), and pets (dogs, cats) act as reservoirs of AMR for first-line and last-resort antimicrobial agents. AMR proliferates in insects, rodents, and pets, and their skin and gut systems. Subsequently, insects, rodents, and pets act as vectors that disseminate AMR to humans via direct contact, human food contamination, and horizontal gene transfer. Thus, insects, rodents, and pets might act as sentinels or bioindicators of AMR. Human health risks are discussed, including those unique to low-income countries. Current evidence on human health risks is largely inferential and based on qualitative data, but comprehensive statistics based on quantitative microbial risk assessment (QMRA) are still lacking. Hence, tracing human health risks of AMR to insects, rodents, and pets, remains a challenge. To safeguard human health, mitigation measures are proposed, based on the one-health approach. Future research should include human health risk analysis using QMRA, and the application of in-silico techniques, genomics, network analysis, and ’big data’ analytical tools to understand the role of household insects, rodents, and pets in the persistence, circulation, and health risks of AMR.

## 1. Introduction

Antimicrobial resistance (AMR) is an emerging global health concern to both animals and humans. In general, AMR is a collective term referring to the capacity of microorganisms (often pathogenic) to develop physical or biochemical mechanisms and processes that render antimicrobial agents ineffective, including antibiotics [1]. Compared to other forms of AMR, antibiotic resistance is most broadly covered by the literature due to its prominence in human and animal health. In this paper, the term antimicrobial resistance is used to refer to both antibiotic resistance and other forms of antimicrobial resistance. Global estimates show that approximately 700,000 deaths occur each year due to antimicrobial-resistant bacterial infections [2]. For example, in 2016 alone, approximately 126,000 people died from extensively drug-resistant and multi-drug-resistant tuberculosis [3,4]. This figure is potentially higher if one considers other antimicrobial-resistant human infections caused by the ESKAPE pathogens [5,6]. ESKAPE refers to a group of multi-drug-resistant pathogens, including—(1) *Enterococcus faecium*, (2) *Staphylococcus aureus*, (3) *Klebsiella pneumoniae*, (4) *Acinetobacter baumannii*, (5) *Pseudomonas aeruginosa*, and (6) *Enterobacter* spp.) [7]. Recently, the list was extended to include; (1) *Clostridium difficile*, (2) *Proteus* spp., and (3) pathogenic *Escherichia coli* [5]. Due to its public and animal health significance, AMR attracted significant global research and public attention [6,7]. Hence, a considerable body of evidence exists on the subject.

The existing literature on AMR, including review papers, focused on the following aspects, among several others—(1) human infections, including nosocomial ones or those associated with healthcare facilities [7,8], (2) occurrence in animal production systems [9], and (3) occurrence, fate, and health risks of AMR in various environmental compartments [10]. To date, AMR was reported in the following environmental compartments—(1) human-impacted soils [11], (2) industrial and municipal wastewater systems [12,13], (3) surface water and groundwater systems, including drinking water systems [14], and (4) ambient environments including particulate matter and aerosols [15,16]. A wide range of antimicrobial-resistant microorganisms and antimicrobial resistance genes of human health concern were detected, including the ESKAPE pathogens [5,6,10].

An increasing body of recent evidence shows that antimicrobial resistance occurs in insects, wildlife, including rodents and birds, and even companion animals or pets [17,18]. Several insects, including edible ones and those associated with household settings such as cockroaches, houseflies, ants, and mosquitoes, were reported to harbor AMR [19,20]. AMR was also reported in household rodents, including mice and rats [21]. In other studies, AMR was reported in common companion animals/pets such as domesticated and stray dogs and cats, as well as wild animals that are often used as pets, such as raccoons [22,23]. The occurrence of AMR in household insects, rodents, and companion animals/pets is a major human health concern. This is because such insects, rodents, and animals might transmit AMR to humans via direct contact or food contamination. However, compared to other compartments such as AMR in humans, and industrial and municipal wastewater systems, comprehensive reviews on AMR in household and edible insects and rodents are still lacking. Barring a few exceptions [24,25], comprehensive reviews on AMR in companion animals/pets are still limited. The few available review papers are mostly limited to developed countries such as the European Community [24]. Thus, AMR in companion animals in developing countries received limited attention. Yet, developing countries have several unique risk factors, including a putatively high number of stray animals and weak and poorly enforced animal and human health regulations.

Therefore, the present review posits that (1) insects and rodents, including edible ones and those associated with household settings and pets, harbor and act as reservoirs of a diverse range of antimicrobial-resistant microorganisms and resistance genes of human health concern; (2) the animal compartments comprise insects, rodents, and pets, which act as transmission vectors and sentinels or bioindicators of AMR; and (3) the persistence and dissemination of AMR via insects, rodents, and companion, pose significant human exposure and health risks. To address these hypotheses, the source-pathway-receptor-impact framework was used to track the occurrence and circulation of AMR in the environment–animal (insects, rodents, pets)–human interface.

Figure 1 presents a summary of the focus of the present paper. Specifically, the various environmental reservoirs of AMR or resistomes that include solid waste and wastewater systems, animal and human health care facilities, livestock production systems, and wildlife, among others. The insect, rodent, and pet resistomes or reservoirs are also shown including the exchange of AMR among them, via various processes, including horizontal gene transfer. These reservoirs or resistomes act as both sources and sinks of AMR, resulting in complex circulation of AMR among the various compartments. Human exposure to AMR via ingestion of contaminated food, inhalation of contaminated particulate matter and ambient air, and direct contact and bites is also shown. Moreover, the occurrence of AMR in pets, rodents, and insects make them ideal sentinels for AMR surveillance. Finally, the human health risks, and future research needs are summarized.

## 2. Antimicrobial Resistance

### 2.1. Nature

AMR is classified into two groups—intrinsic or acquired. While intrinsic AMR results from the ability of the microorganism to survive in the presence of an antimicrobial agent because of an inherent property of the microorganism, acquired AMR emanates either from gene mutation or by the microorganism acquiring extra gene coding for a resistance mechanism [26]. Although antibiotic resistance is commonly studied, others such as antifungal resistance [27,28], antiparasitic resistance [29], and antiviral resistance [30,31] were also documented.

As antimicrobials are commonly used in the management of microbial infections in veterinary and human medicine, and also as growth promoters in livestock, ABR is exacerbated in natural and man-made environments and consequently attracts relatively wider research attention [32,33,34]. Such extensive use and misuse of antimicrobial agents put bacteria under selection pressure to develop resistance [26,29].

The interfaces among humans, animals, and the environment are hotspots for the exchange of antimicrobial-resistant microorganisms and their resistance genes [35]. Previous studies that addressed the impacts of agricultural pollution on AMR showed that interaction with livestock can result in the transmission of antimicrobial resistance to wild animals [36]. The common pathogen exposure pathways of clinically significant AMR are antibiotic usage in humans and livestock, while data on the fate of AMR outside these hosts, is scarce. It is, however, evident that focusing on these compartments only results in an inadequate understanding of the epidemiology of AMR [35]. In this regard, a number of surveillance strategies were used to monitor the occurrence and patterns of AMR in animals and humans [37]. For the human host, these strategies are, however, limited by high costs, ethical requirements, and challenges in sampling protocols. Consequently, a range of sentinels, including houseflies, rodents, shrews, field mice, and chickens, were used for AMR surveillance [35,36,37,38,39,40,41].

### 2.2. Mechanisms of AMR

Antimicrobials are continuously discharged into the environment due to their intensive usage, resulting in the emergence of antimicrobial resistance [42,43]. Microorganisms, including bacteria, can readily adapt and develop resistance against antibiotics, releasing antimicrobial resistance genes (ARGs) into the environment [43]. Through genetic transformation, ARGs are transmitted to environmental microorganisms and pathogens, exacerbating environmental, and human health risks [32]. The development of antimicrobial resistance in microorganisms depends on two key factors, namely, the existence of ARGs and the selective pressure exerted by antibiotic usage [26]. Recently, antibiotic management of common infectious diseases showed diminished effectiveness, owing to prolific use, increased microbial resistance, and the propagation of ARGs via transposable elements and plasmids among microbial communities [43].

Apart from occurrence in livestock and wild animals via interactions through shared environments, AMR is detected in cockroaches [44,45,46], wild mammals such as riverbank voles [36], birds [35], and houseflies [47]. The transmission pathways from these animals to humans are gaining research interest. For instance, studies showed that *Yersinia pestis*, which causes plague, can be transferred from one rodent to another through flea bites [48]. This can cause bubonic plague via rodent-to-human transfer by infected fleas from cats, rats, or wild rodents. Further human-to-human transmission via body fluids can cause the pneumonic plague. The role of wild small mammals as potential reservoirs of AMR is poorly understood, and further research is needed [41]. Another study reported that houseflies carry antibiotic-resistant enterococci, with the potential to horizontally transfer ARGs to other microorganisms [38].

The main mechanisms for the development of antimicrobial resistance related to drug usage are (1) new opportunistic pathogenic microbes emerge, which are usually antimicrobial-resistant; (2) new acquired resistance mechanisms emerge; (3) gene mutation in the host chromosome or plasmid-borne occurs; and (4) due to the dispersion of old resistance genes into new host microorganisms [49]. A more detailed discussion of the occurrence of AMR and its transfer between the environment and humans is provided in earlier reviews [26,50,51]. The problems associated with AMR are additionally convoluted by horizontal gene transfer (HGT), the process through which adaptive genes are exchanged among microorganisms [26]. Incidentally, HGT is important for the evolution of species [49]. HGT in microorganisms proceeds via three key mechanisms—(1) natural transformation, which involves the uptake and integration of DNA derived from the extracellular media; (2) conjugation, a common mechanism for DNA transfer; and (3) transduction, which is bacteriophage facilitated transfer [26].

Acquired resistance is mainly transferred via HGT of environmental or microbial ARGs, and this is facilitated by mobile genetic elements (MGEs) [42]. While intrinsic resistance does not undergo horizontal transfer and is benign in non-pathogenic microorganisms, acquired resistance occurs in some microbial strains, which are often vulnerable to the antimicrobial agent under investigation, and might be horizontally transferred among microorganisms [26]. As a result, several previous studies profiled ARGs and MGEs, such as bacteriophages, insertion sequences, integrons, plasmids, and transposons [26,42,52]. The genes that confer AMR can disperse rapidly, passing on to microorganisms not initially exposed to the selective pressure, thus spreading throughout microbial populations [36]. For example, apart from clinically significant microbial species, many researchers suspect that commensal bacteria such as lactic acid bacteria, enterococci, and streptococci could be reservoirs of ARGs, comparable to those in human pathogens [26,53,54]. The significant health risk related to these bacteria is their propensity to transfer resistance genes to pathogenic microorganisms.

The genetic modifications in acquired resistance alter the defenses of the microorganisms through several pathways. These include (1) altering the target receptor of the antimicrobial agent, (2) altering membrane permeability to antimicrobial agents, (3) enzymic inactivation of the antimicrobial agent, (4) active transportation of the antimicrobial agent via efflux pumps, or (5) by directing metabolic routes to avoid disruption [26]. It is likely that AMR evolved way before the usage of antibiotics for clinical purposes. These resistance genes might arise from the antimicrobial producers that possess resistance genes for self-protection from the antimicrobial products. A further potential source of resistance genes is probably genes whose products participate in the metabolic processes of the microorganisms. Such genes might gradually mutate, which alters the substrate from substrates of biodegradative or biosynthetic routes to antibiotics [26].

### 2.3. Conventional and Emerging Analytical Methods for Antimicrobial Resistance

Following the capture of the host insects or animals such as cockroaches and flies, the microorganisms can be isolated using classical methods (e.g., washing with saline water) and then subjected to standard protocols for antimicrobial susceptibility testing [32,33,39,40,55,56,57]. To identify antibiotic susceptible microorganisms or ARB in a sample, the minimum inhibitory concentrations (MIC) of antibiotics to particular bacteria are predefined [58]. The MIC values are for the defined bacterial species, but they are not defined for many environmental bacteria that do not often grow on culture media. Therefore, these methods are biased in that they focus on selected bacteria based on the isolation media used, while other non-targeted bacteria might also be carrying resistance but are not detected. Additionally, this approach is mainly phenotypic and does not give data on ARGs responsible for the observed resistance. Generally, standardizing culture-based methods provide a simple and inexpensive data comparison, especially in nutrient-starved host matrices.

Cultural methods still dominate studies on AMR. There is a need to improve on methods and implore advanced technologies, such as the sequencing of the whole genome based on antimicrobial-resistant microbial isolates. This could be helpful in providing critical information, e.g., for identifying shared microbial strains, antibiotic resistance genes, and mobile genetic elements [59,60]. Using approaches such as metagenomics in determining the abundance of antimicrobial-resistant microorganisms, resistance genes, and mobile genetic elements, e.g., in the gut of rodents, could provide a clearer and better picture of the routes through which AMR spreads or direction of the transmissions [59,61].

To quantify AMR, it is essential to understand the advantages and disadvantages of conventional culture-based methods and molecular-based methods. Culture-based methods require immediate sample treatment and are limited to particular microbial strains and bacteria that can be cultured on synthetic media [39]. This approach thus excludes bacteria that are in a viable but non-culturable form, or ARG carrying non-target microorganisms that do not grow on the selective media. This can result in false-negative results. The commonly used molecular-based method to detect ARGs is the quantitative polymerase chain reaction (qPCR) method, which is based on the intensification of the nucleic acid [36]. This method involves recovering the DNA of microorganisms from total DNA samples and amplifying ARGs, following standard procedures. Particular microbial genes are determined using predefined primers [37]. The qPCR technique is highly specific, rapid, and has a high accuracy [6,43]. Consequently, through the application of qPCR, the total ARGs in all strains of microorganisms, bacteriophages, mobile DNA fragments such as integrons, plasmids, and transposons, and environmental DNA (eDNA) could be detected. Furthermore, sample processing with propidium monoazide endows qPCR with capabilities to distinguish DNA derived from living cells, as opposed to DNA from dead cells, phages, or fragments of eDNA. Hence, potential ABR donors of ARGs could be enumerated. However, on its own, PCR does not detect the microorganisms carrying the specific resistance genes and infer their significance to human health. Moreover, the method is confined to known genes, rendering it inconclusive on the total resistance to a particular antimicrobial agent. Thus, data for determining a single resistance gene might be inadequate to quantify the actual resistance to the particular antibiotic present. In this regard, highly parallel qPCR analyses, ChipSeq, microarrays, or next-generation sequencing could potentially solve this problem, but they are expensive and usually require high expertise [33,39].

In addition to qPCR, a range of modern techniques are used to detect and quantify ARB. For example, following an extraction, gel electrophoresis was used to determine the quality of the DNA [42]. A similar study used pulsed-field gel electrophoresis to identify DNA from streptomycin-resistant *E. coli* and *Yersinia* strains [48]. Yet another study determined AMR staphylococci using MALDI-TOF [41]. The application of genomic methods in understanding AMR attracted research interest. Extensive genomic studies employing high-throughput sequencing data represent powerful novel approaches that rapidly detect and respond to genetic transformations associated with AMR [62]. Nevertheless, these studies lack mechanistic details. In this regard, computational techniques can speedily and cost-effectively assess the impact of mutations on the function of proteins and their evolution. Recently, methods that depend on mathematical modeling and artificial intelligence were used to predict the occurrence and fate of AMR without the requirement of laborious, expensive, and sometimes inconvenient in-situ analyses [63]. In addition, big data analytics such as artificial neural networks and artificial intelligence already showed great promise in microbiological diagnostic testing and assisting in clinical decision making [64,65]. The main strength of these techniques is their capability to generate and handle large amounts of data and predict the prevalence of AMR from historical data [63,65]. They can, thus, be used to select the best intervention strategies or experimental conditions that generate data for developing preventive and remediation approaches. With the advancement in technology, new analytical methods are expected to emerge. These would cover non-intrusive sampling, minimal sample preparation, and rapid analysis, and are expected to be faster, simpler, and more accurate in the near future.

## 3. The Role of Insects, Rodents, and Pets in AMR Persistence and Transmission

### 3.1. Insects

Several insects, including those associated with household settings, are reported to harbor AMR (Table 1 and Appendix A). To date, AMR in insects was reported in several countries (Figure 2). Table 1 shows that insects harbor microorganisms possessing genes conferring resistance to a wide range of antimicrobials, including ampicillin, tetracycline, streptomycin, ciprofloxacin, gentamicin, sulfamethoxazole, chloramphenicol, penicillin, and kanamycin. Almost all studies use physical traps, sometimes in combination with insecticides, to randomly capture insects present at a location. Houseflies are the most commonly occurring insect in most environments and exist in close association with humans [37,47,66,67]. It is unsurprising, therefore, that houseflies are the most widely studied. These studies use traps that are inefficient at catching all kinds of available insects, which might explain why only a few, most prevalent insects at most sites are the ones studied. For instance, sweeping with a broom was used to catch houseflies in one study [47], while sweep nets were used in another study resulting in a catch of just three types of insects; houseflies (*Musca domestica*), false stable flies (*Muscina stabulans*), and stable flies (*Stomoxys calcitrans*) [67]. However, in the same study, it was unclear whether or not an insecticide was used prior to sweeping.

Insects such as houseflies and cockroaches can move unrestricted between different environmental compartments, allowing them to easily spread antimicrobial-resistant pathogens. This high dispersal ability of insects is set to promote the occurrence and persistence of antimicrobial resistance in humans, emanating in anthropogenic activities in other sectors such as agriculture. For instance, houseflies and cockroaches successfully transmitted antimicrobial-resistant bacteria obtained in pig manure from two commercial confined swine farms [39,51]. In addition, several antimicrobial resistance organisms were isolated from the products of commercial livestock and linked to human health resistance to certain drugs [51,68,69]. In Zimbabwe, chickens were reported to harbor antimicrobial-resistant *E. coli*, exhibiting multi-drug resistance to several antibiotics, including cloxacillin, tetracycline, ampicillin, neomycin, and bacitracin [40]. The study did not investigate the transmission of AMR to humans, other animals, or insects. These findings are critical for decision-and policy-making, particularly in developing countries, where unregulated housing developments close to agricultural facilities are common. Intensive livestock production systems, especially of poultry, pigs, and dairy cattle confined in small areas, involve intensive use of veterinary drugs and antimicrobials in animal feeds. This is a public health concern, as the antimicrobial-resistant pathogens can be transferred from the livestock to humans through meat and eggs and via vectors such as insects and rodents. In the study by Saidi et al. [40], the liver, egg yolk, and heart of the chickens contained antimicrobial-resistant bacteria. In addition, flies and cockroaches can transport the bacteria from the poultry runs to residential homes due to their unrestricted movements.

The alarming rise in antimicrobial resistance, even to the last-resort antibiotics, is an unprecedented challenge to public health, as it limits treatment options of microbial infections. Thus, the World Health Organization has called for a “One Health” approach to find effective solutions to address this challenge [70]. While this approach advocates for collaborative efforts to address this problem in humans, animals, and the environment, specific components might need to be given attention as they might not always be easily classified into the One Health compartment. Examples of such compartments are insects and rodents. Insects and rodents harbor numerous microorganisms, including antibiotic-resistant pathogens, as evidenced in the numerous research findings reported globally, involving different insects sampled from different sources, including households (Figure 1; Appendix A). Some studies reported resistance to last-resort antibiotics like colistin, in bacteria from flies [71,72]. Similarly, the presence of Extended Spectrum Beta-Lactamase-producing microorganisms was reported in recent studies [73]. Understanding how insects acquire these bacteria and transmit them to humans and other animals is crucial to establish effective intervention measures, to prevent the spread of AMR through these often neglected yet important organisms.

Most insects prefer damp areas, and dead and decaying matter, including sewers, carcasses, human and animal fecal matter, and garbage, among others. Through frequenting such unsanitary environments, the insects come into contact with pathogens and subsequently transmit them to other compartments, including humans. The exoskeleton and alimentary canal (or gut) of the studied insects acted as reservoirs for antimicrobial-resistant microbiota, antimicrobial resistance genes, and mobile genetic elements. Besides the exoskeleton and feces, the gut, saliva, or vomit through regurgitation is another additional reservoir and source for transmission [74]. When feeding, houseflies regurgitate liquid from the stomach to dissolve food, then use their sponging mouthparts to suck it up. While houseflies leave fecal spots, cockroaches also leave behind fecal droppings as they feed and move around. About 98% of cockroaches collected in one study had bacteria in both their external surfaces and alimentary canal, though significantly higher for the latter [44]. Using houseflies, the total number of non-fecal-related oocysts dislodged from their exoskeleton was 463, compared to 564 fecal-related oocysts [75]. During the experimental period, the number of non-fecal-related oocysts significantly decreased from 267 to 14 from day 3 to 11, while fecal-related oocysts significantly increased from day 3 to 11 [75]. It was also demonstrated that *P. aeruginosa* could multiply in the gut of cockroaches and be excreted for up to 114 days. These findings show that, in addition to being AMR vectors, houseflies and cockroaches are also amplifiers of antimicrobial-resistant pathogens and their associated resistance genes. However, more work needs to be done to ascertain this phenomenon in detail and confirm its occurrence in other insects and rodents. Pathogens in the guts of houseflies and blow-flies were three times greater than on their body surfaces [66]. These findings confirm that the exoskeleton or external surface of insects and the feces or gut are reservoirs and routes of transmission. The contribution of the exoskeleton is, albeit inferior to that of the gut. Insects have hairs and bristles, as well as a structure called pulvillus with surface adhesive—though on varying parts of their bodies [74,76]. These body parts facilitate the adhesion of substances onto the insect and allow the insect to attach to surfaces as it climbs or walks. Thus, pathogens can adhere to insects, and pathogens attached to these hairs, bristles, or pulvilli might not get dismantled easily, resulting in lower transmission via the external surface compared to the feces [75]. It should be emphasized that more work is needed, which incorporates individual fly assessments to ascertain the location of the pathogens on the insect, as this might assist in disease monitoring, risk assessment, and control, among other epidemiological implications.

In one study, several types of flies (namely, secondary screwworm, *Cochliomyia macellaria* (prevalence-2%), the red-tailed flesh fly, *Sarcophaga haemorrhoidalis* (prevalence-2%), and the black garbage fly, *Ophyra leucostoma* (prevalence-1%)) were sampled using the sweep nets, but the analysis of their guts revealed almost non-existence of the tested pathogens [66]. It also follows that only those insects that are widely associated with disease transmission receive much research attention. These insects include houseflies, cockroaches, and to a less extent, blow-flies. A few earlier review papers investigated the role of houseflies in the transmission of antimicrobial resistance, but these were limited to only antibiotic resistance and bacteria [51,74]. This present review focuses on all kinds of insects and how they act as possible sources of resistant pathogens and ARGs in the environment.

#### 3.1.1. How Insects Acquire Resistant Bacteria

Insects can acquire antimicrobial-resistant microorganisms through various mechanisms. One such mechanism is the physical contact of the flies with polluted environments containing antimicrobial-resistant microorganisms. For example, flies have a hairy proboscis that produces sticky substances, allowing bacteria to attach to these body parts [72]. Hairs on other body parts of flies (e.g., wings, legs) also facilitate the attachment of antimicrobial-resistant microorganisms on their exoskeletons [77].

Insects can also acquire resistant microorganisms by feeding on foods contaminated by these organisms. In an experiment, sterile cockroaches were fed with *Salmonella*-contaminated foods, and the bacterium was isolated in different insect parts, including their guts [78]. This ability of insects to feed on bacteria is exploited in biotechnology. For example, the shift to natural manure led to the search for novel methods to treat manure before application on farms. Many scientists recently reported using housefly larvae to attenuate the antibiotic resistome in swine manure, a practice known as vermicomposting [79,80]. Wang et al. [80] reported that the population of tetracycline-resistant bacteria and their associated tet (M, O, K, W) genes were significantly reduced in manure exposed to larvae of the housefly. Although proven to be a promising biotechnology approach for reducing antimicrobial resistance in manure, the bioaccumulation of antimicrobial resistance genes and antimicrobial-resistant microorganisms in the flies is not given full attention. For example, one study demonstrated that flies could internalize ARB throughout their life cycle [81]. Thus, flies that emerge from such manure treatment practices could be potential environmental reservoirs of ARB and ARGs, and subsequent vehicles for transmission to humans, as it was shown that the housefly could travel as far as 7 km from poultry farms [82].

Apart from directly feeding on food infected with ARB, normal flora of insects can acquire resistance due to exposure to sub-lethal doses of antibiotics. For example, the use of antibiotics in agriculture favored the exposure of many insects to these chemotherapeutic agents, causing selective pressure on the insects’ gut microbiome. A well-known example is the historical use of tetracycline in honeybees, which contributed to the development of AMR in the bees’ gut microbiome [83]. Bacteria in insects can also acquire resistance when these insects feed on specific plants that produce secondary metabolites with antibacterial properties [84].

#### 3.1.2. Insects as Reservoirs of Antimicrobial Resistance

Insects are known to invade and inhabit human foods, whether accidentally or intentionally [85]. Some of these insects occur in habitats such as cracks and crevices within human homes [86]. Their presence in these places could have direct and indirect impacts on human health. Many of these insects were reported to carry large numbers of diverse microorganisms, including antibiotic-resistant ones. For example, in a study conducted in a hospital in the United Kingdom, 82 bacterial strains were isolated from 19,937 different fly species, and 68 of these were resistant to at least one of a set of 11 antibiotics while 11 strains were multi-drug-resistant [87]. The authors also reported bacterial counts as high as 10^10^ colony forming units (CFU) per fly per mL.

Another study demonstrated that the gut of cockroaches represented a favorable environment for the horizontal transfer of resistance genes between *E. coli* and other *Enterobacteriaceae* [88]. The same authors reported the transfer of resistance genes for ampicillin, kanamycin, and tetracycline resistance in the insect’s intestines, thus recognizing cockroaches as potential reservoirs for the transfer of antibiotic-resistant bacteria. Yet another study reported that houseflies were more than just mechanical vectors of AMR, but that their intestines were a favorable environment for the transfer of ARGs to closely related bacterial species [89]. The same authors reported that plasmid-borne cephalosporin resistance genes were successfully transferred from *E. coli* to *Acinetobacter* sp. and *Pseudomonas fluorescence* within the intestines of the flies. In a similar study, one-day-old houseflies were fed with an antibiotic-resistant *E. coli* suspension, followed up to the next-generation [81]. The results showed that, apart from horizontal gene transfer within the fly, vertical transfer was possible from one fly generation to another. In addition, the antimicrobial-resistant bacteria were transferred to the eggs, larvae, pupae, and finally, the next-generation of adult flies. Related studies reporting on horizontal gene transfer within the fly intestine include the transfer of tetracycline genes among *E. faecalis* [90] and ampicillin-rifampicin resistance transfer in *E. coli* [91].

Most insects captured and studied in most studies harbored antimicrobial-resistant microbiota, ARGs, and MGEs [39,51,58]. The feces of the German cockroach (*Blattella germanica*) and the digestive tract of houseflies were dominated by the multi-drug-resistant bacteria, *E. faecalis*, along with the associated ARGs; tet(M), erm(B), gelE, esp, asa1, and the Tn*916/1545* transposon family obtained from swine farm manure [39]. Metagenomic analysis revealed the simultaneous occurrence of pathogens, with ARGs and MGEs responsible for horizontal gene transfer among bacteria in sludge samples of domestic wastewater in Beijing, China [42]. Hence, insects that breed or feed or inhabit in such media as wastewater or fecal material potentially carry the ARMs, ARGs, and MGEs along with them, impart resistance to other microbes, as well as pass them on to humans, as they get in contact with human skin and foodstuffs with their pathogen-laden saliva, exoskeleton, and fecal matter.

#### 3.1.3. Insects as Vectors of AMR

The antimicrobial-resistant microorganisms carried on insects can be transmitted to other insects and surfaces through different mechanisms. This transmission could have severe public health implications. For example, a study conducted in South Africa reported no differences in the bacteria isolated from cockroaches and patients in a hospital, during an outbreak [92]. The authors concluded that cockroaches played a role in spreading ESBL-producing *K. pneumoniae* in neonates within the studied hospital environment. This conclusion was further strengthened when the number of new cases dropped significantly, following intervention measures that resulted in eliminating the insects. Similarly, a direct correlation was observed between the density of flies and the incidence of Shigella infections in Bangladesh [93].

On the one hand, the ARB occurring on the outside layers of the insects, usually on the legs and mouth, are transferred mechanically as the insects interact with different types of surfaces and other insects. The mechanical transfer was reported in a study that demonstrated that *E. coli* carried on the outer surfaces of flies was transferred to different food types, including steak, milk, and potato salad [94]. On the other hand, the internally-borne bacteria are transmitted through the insects’ feces, saliva during feeding (regurgitation), or during the decay of dead insects [95,96]. In one study, houseflies were fed with fluorescent-labeled bacteria to investigate the migration of the bacteria in the fly and study the possibility of transmission to other surfaces following ingestion [97]. It was observed that the fly’s crop presented a favorable environment for their thriving and that the bacteria were present in water and food with which the flies had contact. A study on cockroaches fed with an antibiotic-resistant strain of *Salmonella* reported the ARB were transferred to nymph grown under sterile conditions [78]. It was observed that both the *Salmonella*-fed cockroaches and the sterile ones harbored the test organism. This study demonstrated that cockroaches (and other insects) could pick up bacteria (including antimicrobial-resistant ones) from dirty environments and transfer them to clean environments and other insects. Allen [98] demonstrated the transfer of ARB from cockroaches through defecation, using a laboratory experiment. In this study, cockroaches were fed with human saliva containing *Mycobacterium tuberculosis*. The feces from the insects were then collected and examined for the presence of the bacilli, and it was observed that the bacterium was present in the insect’s feces for up to eight weeks, at room temperatures. Using metagenomics, it was also demonstrated that the intestinal microbiome of cockroaches shared up to 90% similarity with that of the insects’ feces [99], further supporting the potential of the insects as vectors for ARB. Conversely, AMR can be transmitted from humans to animals, and from insects such as flies and cockroaches, from human hospitals to animals. This complex and continuous exchange of AMR among the various compartments or resistomes, which also applies to rodents and pets, underpins the ‘one world, one health’ concept.

In addition to sharing their habitats with humans, some insects serve as food, due to their high energy content. With a global shift to insects as alternative protein sources [100], humans could get infected by eating inadequately cooked or raw insects. Thus, as described for the transfer of resistant bacteria by insects to different surfaces and food, the same mechanisms apply to the transfer of these bacteria to humans. Humans can get infected with insect-borne ARB by ingesting insects. In addition, transmission might also occur indirectly through contact with surfaces and kitchen utensils infected by these flies [101], or through consumption of food or drinking water contaminated by these insects.

One laboratory study used nalidixic acid-resistant *Salmonella* spp. to ascertain whether cockroaches could acquire ARB from contaminated sites and subsequently transmit these to clean surfaces, water, and other cockroaches [102]. The results showed that the infected cockroaches successfully transferred the resistant bacterium into water, table eggs (whole uncracked egg surfaces), and other insects, with contamination occurring up to four days following the start of the experiments. For biting insects like spiders [103], mosquitoes [104], and bed bugs [105], the transfer of ARB to humans could occur through insect bites.

Earlier literature suggests that effective management for the control of the spread of AMR through insects should target the reduction of insect populations [39]. Given that flies, for example, can travel for long distances, animal production facilities must be situated far away from human settlements. However, rapid industrial development in many developed countries led to intensive animal farming in residential areas [106]. Therefore, measures that aim to prevent the escape of flies from these settings should be put in place. This could be done using fly bait areas that would attract the insects and prevent them from moving out of the farms. The possibility of AMR transfer by insects from animal farms to humans is a major challenge for most low- and middle-income countries, especially in sub-Saharan Africa, where animal farms are usually located in proximity to houses [107]. Such closeness would allow for the easy movement of insects between the farms and households, thus exposing humans to ARB.

Despite the numerous studies on the role of insects as reservoirs and vectors of AMR transmission, a number of gaps exist. Most studies were carried out on houseflies and cockroaches, and only a few studies looked at other insects such as spiders, butterflies, mosquitoes, moths, for example, with which humans regularly come into contact with in households, gardens, and recreational facilities like parks. This is particularly crucial for resource-limited settings in sub-Sahara African countries, where the tropical climate promotes the proliferation of insects, resulting in regular exposure to other insects, such as ants, beetles, grasshoppers, and spiders.

Studies on AMR in insects mostly focus on understanding the prevalence and potential transfer to humans and other animals. However, given that most resistant organisms present in insects are acquired from human-impacted environments, insects could be used as sentinels for studying the resistome circulating within a community. The control of insect populations is mostly carried out with insecticides that contain heavy metals, which co-select for AMR [108,109,110]. A plasmid was identified in *Bacillus thuringiensis*, carrying antimicrobial and insecticidal resistance genes [111]. Thus, the co-selection of insecticides for AMR in insects requires further investigation.

Previous studies are mostly on antibiotic resistance associated with antibiotics and bacteria. Little information exists on the potential of insects to act as reservoirs of drug-resistant viruses and fungi. Another critical gap for establishing the risk associated with ARB and ARGs in household insects is the lack of information regarding the transfer efficiency of the insect bacteria to different environments (e.g., water, household surfaces, and foods). To obtain such information, it is necessary to understand the proportion of resistant bacteria within different insects. In addition, how long the insect would have to be in a contaminated environment to carry a substantial number of bacteria and how much contact time would be necessary to deposit substantial amounts of these bacteria to different surfaces, need to be investigated further.

#### 3.1.4. Insects as Sentinels of AMR

As insects are potential vectors of human diseases, the presence of pathogens in insects is of importance and should be explored further. Research shows with some certainty that insects can be a source of AMR [51]. For instance, the guts of insects are reservoirs of AMR, especially ARGs and this AMR can be transmitted between different habitats, including those mostly frequented by humans [51]. In particular, household insects such as cockroaches and houseflies, which are highly associated with food animals, can potentially facilitate the transfer of AMR between farms and urban settlements. There is research evidence proving that these insects (cockroaches and houseflies) harbor some multiple drug-resistant bacteria, that there is horizontal transfer of antimicrobial genes in these insects, and that they can potentially transmit AMR from one environment to the other [38]. Additionally, several household insects such as bed bugs and fleas also harbor resistant microorganisms containing ARGs [58,112,113]. In the gut of fleas, HGT was reported from *E. coli* bacteria to *Yersinia pestis*, resulting in the antibiotic-resistant *Y. pestis* from patients in Madagascar [112,113].

As mentioned earlier, houseflies belonging to the families *Muscidae* and *Calliphoridae* can transmit microbial organisms, which besides bacteria also include fungi and even viruses [74,114,115,116,117,118]. In addition, there is evidence that some of these transmitted microorganisms are resistant to antimicrobials [119,120,121]. These flies normally prefer in-house dwellings and therefore associate with humans on a large scale [122]. They use decaying matter and excrement for laying eggs and for their nutrition [123]. Their nature, therefore, involves the transmission of microorganisms through fecal–oral transmission [74]. Their mobility from one habitat to the other even over quite long distances and their ability to disseminate pathogens such as bacteria, which they carry on the surface of their exoskeleton and alimentary canal, supports the idea that they also spread AMR to other organisms, including humans [93,124]. This is more so because some studies highlight the importance of the gut of the fly in providing an excellent environment for the HGT of AMR genes and the transport of AMR microorganisms [89,90]. In households, flies enhance the spread of pathogens and AMR to humans, generally through food contamination, when they are in direct contact with the food. More specifically, the transmission of AMR bacteria can be through regurgitation, transfer from the exoskeleton, excretion, and vomiting [95].

Houseflies are suggested as good candidates to use as sentinels in the surveillance of prevalence and spatial distribution of AMR in microbiota, such as bacteria in humans and animals, for several reasons. Houseflies have unrestricted movement such that they commonly occur on or around humans and animals, are easy and cheap to catch using simple traps, and are not subject to ethical issues, as with people [37]. These flies are also able to access areas not easily accessible, and thus can acquire AMR organisms and their pathogens from their diverse niches. Using a 16S rRNA-based microbial metagenomic analysis, ARGs and MGEs detected in the tested flies were similar to those observed in the studied sympatric animals, especially poultry feces [37]. However, robust isolation and fingerprinting protocols are needed before flies can be deemed sentinels for AMR and MGEs in microbiota linking to specific animals or to humans. This is because flies travel and get in contact with a myriad of media and environments, such that it becomes difficult to use them as sentinels of AMR for a particular population. Cockroaches were also shown to spread ARB from swine farms and might also be used in studies to establish if they can also be used for surveillance purposes as sentinels of antimicrobial resistance in humans and animals. Continuous monitoring of antimicrobial resistance is vital, and the use of easier and cheaper ways in such studies is imperative.

Just as with houseflies, cockroaches are also known to spread AMR microorganisms such as bacteria, viruses, helminths, and protozoan, normally through food contamination in house dwellings [125,126,127]. In a house, cockroaches normally occur in kitchens, food pantries, and bathrooms and can easily move from one room to the other, in the process carrying and transmitting microorganisms, some of which are of concern, in terms of public health [128]. A combination of intimate interactions with human dwellings, including stored food and contact with waste such as sewage, garbage, and other environmental wastes, most likely impact the spread of AMR from cockroaches to humans [129]. *Staphylococcus aureus* isolated from cockroaches sampled from houses and restaurants was shown to exhibit AMR, and this poses a health risk as the antimicrobial-resistant *S. aureus* can potentially be easily transmitted to humans from the cockroaches [130,131,132].

Edible insects, which form part of some human diets, also present a cause for concern when it comes to the spread and transmission of AMR in households. Although there is not much research done on AMR in this area, a few studies showed that some edible insects containing different microorganisms in their guts contribute to the distribution of ARGs [133,134,135,136]. Roughly about two billion people, mostly from Africa, Asia, and South America, eat insects as part of their diets as they are a source of high-quality nutrition, and evidence shows that the edible insect sector is growing swiftly [137,138]. Although several studies showed microbiological risks associated with edible insects, as evidenced by their insects’ microbiota [139,140,141,142], a lot still needs to be done in this regard, and it is important to assess the safety of edible insects. However, existing evidence indicates the emergence and spread of ARM and ARGs facilitated by edible insects to humans.

There are increased concerns with the mass breeding and continued release of genetically modified (GM) insects, such as mosquitoes and agricultural pests. As these genetically modified insects are bred, the antibiotic tetracycline is normally added to their feed, and overtime, these can develop AMR in their microbiota [143]. The main concern is that when these GM insects are released to the environment, they can then potentially carry and transmit antibiotic resistance to naturally occurring insects, in such habitats as well as humans, which are more than often associated with such insects in their homes and farms [143]. For example, transfer of this AMR to humans can be through ingestion of the larvae of genetically modified mosquito in contaminated water or the ingestion of the genetically modified pests at the larval stage, when they can normally contaminate vegetables and fruits.

Overall, in the case of AMR in insects, previous studies mostly focused on insects of economic interest such as agricultural insects, bees, and those insects normally associated with pathogens of medical interest such as flies and cockroaches. Insects that do not necessarily fit in these categories are somewhat overlooked, yet they might be significant reservoirs, vectors, and transmitters of AMR.

### 3.2. Rodents

#### 3.2.1. Rodents as Reservoirs of AMR

Besides insects and parasites [148,149], rodents such as rats, mice, and even edible ones share the same environment with people, creating high chances of transmission and exchange of the AMR [150]. A classic example is that of household rodents that frequently invade peoples’ homes and therefore closely interact with humans. In instances where small mammals such as household rodents harbor resistant microorganisms, the transmission of these microorganisms to humans is highly probable via different pathways, such as direct interaction, through the food chain where rodents form part of the household diet or rodents come into direct contact with food, as well as through contact with surfaces. This can then result in the emergence of diseases that are not easy to manage [151].

The significance of rodents, including household rodents, as a reservoir of AMR, is inadequately recognized, with barely any large-scale research on this, as compared, for example, to AMR in humans. However, these small mammals play a crucial role as sentinels, reservoirs, and vectors of AMR, especially where anthropogenic factors such as pollution are involved or in one way or the other they associate with human waste [152,153,154,155]. Figure 3 shows the global distribution of studies reporting AMR in rodents based on the literature retrieved from PubMed Central and Google Scholar databases. Comparing Figure 3 to Figure 2 shows that fewer studies exist on AMR in rodents than pets. In Africa, the studies on ARM in rodents are even fewer than those in insects. In summary, microorganisms in rodents harbor genes conferring resistance to ampicillin, penicillin, amoxicillin/clavulanic acid, tetracycline, streptomycin, co-trimoxazole, trimethoprim, ciprofloxacin, cefotaxime, gentamicin, apramycin, sulfamethoxazole, chloramphenicol, cephradine, cefuroxime, nalidixic acid, amoxicillin chloramphenicol, and orfloxacin (Table 2). Bacteria is one of the common groups of antimicrobial-resistant pathogens harbored by these rodents [156,157,158,159,160], some of which are of serious concern to human health, as they can be transmitted together with the AMR determinants in the shared household environment. Examples of some bacteria species isolated from rodents include *Yersinia pestis*, *Salmonella* Typhimurium serotype, *Streptococcus moniliformis*, *E. coli*, and *Yersinia enterocolitica*, some of which were observed to be resistant to antimicrobials and yet they are responsible for causing different types of diseases in humans [160,161,162,163]. Therefore, one can consider rodents as multiplication foci for ARGs and are thus sources of AMR, which can be transmitted to other organisms such as humans, in the case of household rodents [178]. Evidence from previous research showed that some rats sampled from homes as well as within shrubs surrounding homes, exhibit AMR, making it easy for such resistance to be transferred from the rats to humans due to interactions [178].

In other examples, rodent species such the house mouse (*Mus musculus*) that flourish in urban areas including human dwellings, pose a huge health risk to other species including humans, as they contaminate the environment with pathogenic microbes and in the process vectoring and transmitting any form of AMR organisms that they carry. The house mouse, commonly found indoors where it associates with humans, was reported to carry some pathogenic organisms such as *C. difficile* [164], *Campylobacter* [165], and *Salmonella* [166], which poses a huge threat if some of these bacteria exhibit AMR. A study reported multiple AMR genes in house mice, and interestingly, the male mice exhibited more AMR genes compared to the females [167]. This is attributable to the fact that male mice normally have a bigger home territory compared to females, and it also increases their exposure to any AMR microorganisms [168]. AMR in rodents is not peculiar to those that were exposed to some form of antimicrobial agents, for example, antibiotics. In fact, it is also prevalent in some rodent populations that were exposed to antimicrobials [159]. This suggests that in some instances, AMR might occur naturally in some microbial populations [169,170,178].

Overall, the AMR phenomenon in rodents, among other small mammals, is complicated, and the determining factors that contribute to AMR in these animals are not always clear cut and distinct. Reports on different carriage rates of AMR in rodents, including household rodents [154,155,171] and the several reports demonstrating that AMR in rodents is largely determined by different anthropogenic factors [154,155,172], are examples that indicate how AMR is complicated in these animals. Given such complexity of AMR in animals, it calls for more detailed and more specific research on the subject, to clearly determine and understand factors that contribute to AMR in such species. Such research can present a platform on which to base mitigation measures against AMR.

#### 3.2.2. Rodents as Vectors of AMR Transmission

The spread of AMR is facilitated through the transfer of resistant microbial organisms between different organisms, including transfer from animals to humans or vice versa [173]. The mechanisms that control the successful transfer and transmission of resistant organisms and their genes from rodents to humans are not very distinct and clear. However, the intimate interactions between rodents and humans in households and farms, among other areas, as well as their high mobility, mean rodents most likely vector and transmit antimicrobial-resistant microorganisms to humans through contact of fecal materials and then contamination of human food and drinking water [160,174,175,178]. For example, invasion of agricultural land by rodents is a potential threat in the spread and transmission of AMR to humans through contamination of livestock that is utilized by humans for food as well as any other food contamination, such as grain that is also utilized by humans for food [61]. The house mice have more intimate contact with humans as they have found a niche position in structures utilized by humans, such as homes, schools, and restaurants. This, along with other urban rodents, was proved to harbor antimicrobial-resistant bacteria such as *E. coli* [156] and *S. aureus* [157], which can easily be transmitted to humans via—(1) direct or close contact, and (2) contamination of utensils and food. This can be exacerbated by non-hygienic handling of the food.

In the food chain, rodents are normally one of the initial links close to the bottom of the chain and therefore can easily become a microbiological transmitter for several predators, including humans. In several parts of the world, including Africa and Asia, wild rodents form part of the traditional household diet, and if such rodents are AMR contaminated and then ingested, this might easily become a pathway through which AMR-resistant organisms carried by these rodents can be transmitted to humans. Wild rodents such as riverbank voles and wood mice harbor antimicrobial-resistant bacteria (e.g., *Enterobacteriaceae*) [159], and if these form part of the household diet, such resistant bacteria can probably be transmitted to humans through ingestion. Current concerns include AMR transmission from rodents to humans, especially focusing on resistant microorganisms that are known to enter the food chain, causing ease of transmission from animals to humans.

Indirectly, rodents and even pets might harbor parasites possessing antimicrobial resistance, which might be transferred to humans via such parasites. A typical example is the emergence of antibiotic-resistant *Yestinia pestis* in bubonic plague patients in Madagascar, which was linked to horizontal gene transfer from rat fleas [48]. The role of parasites in the transfer of AMR from rodents and pets to humans and vice versa further demonstrated the complexity of AMR. Yet, this aspect is relatively less studied compared to the direct role of rodents and pets in the persistence and transmission of AMR.

#### 3.2.3. Rodents as Sentinels of AMR

The capacity of rodents to act as reservoirs and vectors of AMR points to their potential use as sentinels of AMR occurrence, transmission, and potential human health risks. Sentinels or bioindicators are organisms with the potential to be used as an early warning system for human health risks. A few studies showed that small animals, including mice, voles, and insectivores, are effectively used as sentinels of antimicrobial-resistant microorganisms and their ARGs [36,41]. In this regard, the detection of various AMR in rodents might guide further investigations of AMR and humans and potential health risks. Although there are several forms of AMR and ARGs, namely antibiotics resistance, antifungal resistance, antivirals resistance, and antiparasitic resistance [29], the type of AMR and ARGs often studied is antibiotic resistance. Several studies and reviews concerning AMR, especially veterinary AMR, typically focus on antibiotic resistance, while excluding other forms of antimicrobial resistance. However, this does not mean that other types of AMR besides anti-bacterial resistance are less important, as they can potentially lead to devastating consequences. Therefore, further work is needed to understand the potential of other rodents, including those hunted for food. Such research should extend beyond antibiotic resistance to include other forms of antimicrobial resistance (e.g., resistance to antifungals, antivirals). Compared to humans, AMR in wildlife, including rodents, is significantly less studied; hence several questions still need to be addressed pertaining to AMR in wildlife and the implication on human health [59]. There is, therefore, an urgent need to carry out studies that focus on the human–animal transmissions of AMR and to clearly identify and document original sources and ranges of AMR in animals such as rodents and the transmission pathways.

### 3.3. Companion Animals/Pets

#### 3.3.1. Pets as Reservoirs of AMR

Pets or companion animals are domesticated animals normally kept within or near a household for purposes of companionship. These include dogs, cats, rabbits, birds, horses, rats, rabbits, guinea pigs, fish, snakes, and amphibians. Other reasons for pet ownership include safety, exercise, and as service animals trained to carry out specific tasks for the benefit of an individual, with a physical, mental, or sensory disability. Incidentally, owning pets can also reduce alienation, particularly in contemporary high-tech urban societies, which resulted in reduced human physical interaction. Medical evidence shows that pets can improve coping with cardiovascular diseases, can reduce depression [179,180,181], and can enhance child emotional and social development [182,183,184]. The benefits of owning pets led to widespread and increasing pet ownership. Recent surveys showed that approximately 62% of US and 85% of European households own at least one pet [185,186].

Pets are susceptible to many diseases such as urinary tract infections, post-operative wound infections, dental infections, and respiratory tract infections that might require the use of antimicrobials [187,188,189]. Figure 4 shows the global distribution of studies on AMR in pets based on information retrieved from PubMed Central and Google Scholar databases. Table 3 presents common pet infections, including those resistant to antimicrobials. Compared to other animals, pets are more likely to be given a higher level of healthcare because of their emotional and social proximity to their owners. This has led to the high usage of antimicrobials to treat diseases and wounds. In turn, the widespread use of antibiotics led to an increasing prevalence of multi-drug resistance among canine and cat bacterial pathogens. Susceptibility tests of bacterial isolates from UTI in dogs showed multi-drug resistance [188,190,191]. Microorganisms in pets were reported to harbor resistance to first-line and last-resort antimicrobials, including ampicillin, imipenem, colistin, methicillin, cefotaxime, ceftazidime, cefazolin, chloramphenicol, tetracycline, doxycycline, co-trimoxazole, cefotaxime, amoxicillin, lincomycin, enrofloxacin, ofloxacin, ciprofloxacin, amoxicillin/clavulanic acid, erythromycin, teicoplanin, vancomycin, cephalosporin, gentamycin, and tobramycin (Table 4). Some studies demonstrated that antibiotic resistance in pets increase over time [192]. For example, in China, multidrug-resistant bacterial isolations in China increased from 67% to 75% between 2012–2017 [193]. An even more worrying scenario that raised public health concern is a high usage of human critically important antibiotics (CIAs), including macrolides, (fluoro)quinolones, and 3rd and 4th generation cephalosporins on dogs, cats, and horses [194]. A recent survey showed that CIAs were mostly used for treating urinary tract diseases in cats (62%) and dental disease in dogs (36%) [194]. The regular use of antibiotics increases incidences of resistance; hence, animals that frequented treatment had more resistant *S. aureus* compared to healthy animals [195]. Resistance to vancomycin, an important alternative to methicillin-resistant bacteria, is increasing. Despite vancomycin recording a very low resistance of up to 0% [196], recent studies showed increasing vancomycin resistance as high as 24% [197]. Moreover, antimicrobial resistance among bacterial species that are opportunistic to humans, such as *Klebsiella* and *Acinetobacter* spp., is an emerging human health concern [198,199,200,201].

In one study, all isolates of *Staphylococcus* spp. from ear canals (otitis externa) of pet dogs were resistant to at least one antibiotic, while 77.1% of *Staphylococcus* isolates were multi-drug-resistant [191]. The least effective antibiotics were the penicillins, ampicillin, amoxicillin, and amoxicillin-clavulanic acid [191,210]. *E. coli* isolates from UTI were most resistant to oxytetracycline and ampicillin [211]. In another study, microbial isolates from dog UTI showed decreasing effectiveness of enrofloxacin, cephalexin, and oxytetracycline and increasing AMR in *P. aeruginosa* and *E. faecalis*, over ten years (1999–2009) [202].

#### 3.3.2. Pets as Vectors and Sentinels of AMR

Companion animals can transmit AMR bacteria to humans through direct contact or indirectly in their shared environment. Pets transmit several bacterial infections to humans via saliva, urine, feces, aerosols, and are major sources of zoonotic diseases. Some of the common pathogens causing bacterial infections from dogs include *Pasteurella, Salmonella, Brucella, Yersinia enterocolitica, Campylobacter, Capnocytophaga, Bordetella bronchiseptica, Coxiella burnetii, Leptospira, Staphylococcus intermedius*, and Methicillin-resistant *S. aureus* [212]. AMR was reported in these pathogens, pointing to the possibility for AMR transfer to humans via pets. Humans can acquire infectious antimicrobial-resistant bacteria from pets via (1) direct transmission of resistant bacteria to humans, or (2) the transfer of ARGs from non-infectious bacteria (e.g., some *Enterococcus* spp.). The possibility of resistant bacteria passing resistance genes to microbial flora endemic to humans has long been a concern [213]. Although *Enterococcus* spp. are considered endemic bacteria to the gut of humans and pets, they can act as reservoirs of AMR genes that can be transmitted to pathogenic bacteria such as *S. aureus* and *Streptococcus* spp. [214]. However, there is limited direct evidence linking AMR in pets to subsequent human exposure and health risks. Clearly, multi-drug resistance in pets is a public health concern, and wide-ranging and systematic mitigation measures are required (Section 5). Despite the evidence that antibiotic usage promotes the selection of bacteria with AMR [215,216,217], one study reported no association between antibiotic usage and resistance of *E. coli* in dogs and cats [24]. This seems to suggest that the mere use of antibiotics is not the main driver of acquired resistance; instead, it is a result of the quality and frequency of usage.

Due to their closeness and direct contact with humans, a high risk exists for the transmission of AMR from pets to humans (and vice-versa). Human–pet interactions also occur via saliva, urine, feces, aerosols, and direct dermal contact, further increasing the risk of AMR transmission. Dogs and cats are among the most studied pets, but there is a need to understand the occurrence and potential transfer of AMR to humans via other under-studied pet animals (e.g., birds, reptiles).

## 4. Human Exposure and Health Risks

### 4.1. Human Exposure Pathways

The identification and estimation of the adverse human health effects of AMR, including disease incidences and outbreaks, are challenging [222]. This is partly due to a combination of methodological limitations and the lack of essential information for determining risk. Figure 1 presents the circulation and major human exposure and transmission pathways of antimicrobial resistance from insects, rodents, and household pets, along with the source–pathway–receptor–impact continuum. In summary, human exposure to AMR in rodents, insects, and pets might occur via multiple pathways—(1) direct skin or dermal contact, especially in the case of pets (e.g., through licking of pets), and to a lesser extent, household rodents and insects, (2) ingestion of human food including drinking water cross-contaminated by rodents, insects, and pets harboring AMR, (3) consumption of edible rodents and insect, and associated foods contaminated with AMR, especially poorly cooked food, and (4) inhalation of AMR in ambient air and particulate matter contaminated by rodents, insects, and pets (Figure 1). Although not traced to rodents, insects, and pets, the role of airborne transmission of AMR, and the related MGEs, transposons, and integrons, tnpA, and intl1 is well established [15]. In turn, humans harboring AMR originating from rodents, insects, and pets might further disseminate it into the environment, family members, and the community [223]. However, empirical evidence on the role and relative contribution of the various pathways of human exposure to antimicrobial resistance elements is still lacking.

To date, most review studies on general antimicrobial-resistant microorganisms and exposure to humans overlooked the role played by flies, rodents, or household animals or pets in transmission and risk to human health [224,225,226]. Dogs, as well as other sympatric animals such as pigs and chickens, were shown to share their multidrug-resistant bacteria and ARGs with the studied flies and rodents in the cases of HGT [37,39,144]. For example, using a 16S rRNA-based microbial analysis, all bacterial phyla identified in the caught flies were also present in dogs and the other animals studied [37]. In turn, the rodents and flies that generally live in close association with humans are most likely to transfer the ARGs and MGEs to humans via the pathways summarized above. However, further studies should investigate the contribution of the above-mentioned points of human exposure to antimicrobial-resistant microorganisms and genes.

In one study, it was suspected that an antibiotic-resistant *Yersinia pestis* isolated from patients in Madagascar was a result of the transmission of these bacteria through infected fleas to humans. It was inferred that there was HGT of AMR from *E. coli* bacteria to *Yersinia pestis* within the gut of fleas, which in turn then transmitted the antibiotic-resistant *Y. pestis* to humans [52,113]. In another study, AMR was reported in bacteria isolated from cockroaches in nursing homes and long-term care facilities in Taiwan [45]. In another study, cockroaches, blow-flies, and houseflies were observed to be highly contaminated with human intestinal and food-borne bacterial pathogens in human homes [47,66]. These studies did not ascertain whether or not the isolated pathogens were resistant to antimicrobials or carried ARGs and MGEs. However, it was widely accepted in all these studies that insects are vectors of human pathogens that cause a number of human infections worldwide when they come into contact with food and utensils. The role of insects as vectors of foodborne pathogens was investigated on the assumption that the presence of food-borne pathogens in the guts and external surfaces of the flies caught in the bins of restaurants imply that these flies are capable of carrying pathogens and contaminate food or food-re contaminate food or food-related utensils for humans.

It can be inferred that the insects are also capable of transferring antimicrobial-resistant pathogens, their genes, and MGEs, if present, to humans. This is because some of the identified pathogens in the above studies harbored AMR traits to several antimicrobials [40,227]. For example, stored product insects such as the red and confused flour beetles (*Tribolium castaneum*, Herbst, and *Tribolium confusum*, Jacquelin duVal, respectively) had antibiotic-resistant and low concentrations of virulent enterococci in their guts and were shown to harbor MGEs, Tn*916/1545* family of conjugative transposons that commonly host the ermB and tetM genes [145]. In controlled laboratory experiments, the red flour beetle was shown to transmit *E. faecalis* OG1RF—pCF10 strain with antibiotic resistance from contaminated poultry and cattle feed to sterile feed [146]. In one of the few studies on vector–human transmission, identical *Enterobacteriaceae* resistant to extended-spectrum cephalosporins were detected in dogs from households whose owners had a rectal culture positive for extended-spectrum cephalosporin-resistant *Enterobacteriaceae* [228]. Comprehensive insect/household pet to human transmission studies are scarce and so are recommended to ascertain the role of flies, rodents, and pets as vectors and sentinels of antimicrobial resistance organisms, ARGs, and MGEs to humans. Such studies should also encompass strict or controlled infection measures, given that flies, insects, and pets are mobile and traverse varied environments from where they can carry antimicrobial-resistant microorganisms and ARGs. Further, it is largely unknown to what extent the resistant microorganisms occurring in the insects, rodents, or household pets studied present a risk to humans in terms of resistance to certain antibiotics used for humans.

### 4.2. A Summary of the Inferential Evidence Pointing to Potential Human Health Risks

Despite the lack of strong evidence, the emergence and spread of AMR pose potential human health risks. Multi-drug-resistant human pathogens are one of the major causes of morbidity and mortality in patients undergoing some challenging surgeries, such as hip replacements and organ transplants [229,230]. AMR in humans might result in the following adverse health outcomes—(1) treatments normally administered are rendered ineffective, and (2) infection control becomes difficult, increasing the spread of infections and even death, compared to infection with non-resistance pathogens. Nevertheless, human health risks also vary due to differences in sensitivity. For example, in Zimbabwe, about 50% of the human immuno-deficiency virus (HIV) seropositive patients were reported to have significantly reduced susceptibility and high-level resistance to penicillin compared to only 16% in HIV seronegative patients [231]. Babies and children are especially vulnerable given their reckless behavior, and poor hygiene, consequently, mere landing and brief skin contact with insects might result in dislodging of pathogens and associated ARGs from their exoskeleton. However, research is scant on the effect of sensitivity in humans of varying ages or health conditions, among other relevant factors. Besides questionnaire or interview surveys focusing on limited factors, data on the actual human health risk posed by antimicrobial resistance determined using quantitative microbial risk assessment (QMRA) are still lacking. Previous studies assumed human health risks based on exposure to risk factors such as poor hygiene [45,232]. Farmworkers exposed to antimicrobial resistance microbiota from livestock were infected with the antimicrobial-resistant microorganisms, as they are in close contact with the treated animals [232,233]. In the same vein, humans in close contact with their household pets are also likely to get infected, resulting in person-to-person transmission of AMR. Consumption of raw or inappropriately processed foods with no prior preservation adds to public health risk as this might harbor live bacterial cells capable of transmitting AMR to humans. This is because food processes that kill bacteria and ARGs in food products decrease the risk of transmission of AMR, hence reducing the gene pool for AMR traits [225].

The occurrence of AMR in rodents and insects, including edible ones, and pets, raised human health concerns. Yet, evidence directly linking such AMR to human exposure and health risks remains weak. Here, selected studies pointing to potential human health risks are summarized. For example, the antimicrobial-resistant *Salmonella* strain was isolated from two rodent meat and food products (*Rattus tanezumi* and *Bandicota indica*) from Thailand markets [234]. However, based on the results, it was unclear if this *Salmonella* was acquired from the live rodents or handling and processing of the rodent meat [234]. Evidence drawn from Thailand and the Netherlands shows the existence of transferable ARGs in specimens of edible insects (small cricket powder, small crickets, locusts, mealworm larvae, giant waterbugs, black ants, winged termite alates, rhino beetles, mole crickets, silkworm pupae, and black scorpions) from Thailand and Netherlands [135]. In another study, ARGs were also reported in fresh edible insects from the Netherlands and Belgium, particularly in crickets and mealworms [133]. Similarly, transferable ARGs were also detected in edible grasshoppers [134].

Most edible rodents and insects reported in the literature are also widely consumed in Africa. However, similar to other regions, the link with human health risks is still poorly understood, highlighting the need for further research. The consumption of edible rodents and insects is putatively high in developing regions (e.g., Africa, Asia, and South America), especially among low-income households who cannot afford meat. Therefore, these regions might represent ideal sites to understand the potential human exposure and health risks associated with the consumption of edible rodents and insects with AMR. Therefore, comprehensive studies are required to understand the occurrence of AMR in raw or uncooked rodents and insects and the impacts of handling and culinary practices on AMR at the point of consumption.

The lack of compelling evidence on the human exposure and health risks associated with AMR in rodents, insects, and pets might reflect the methodological limitations of existing studies predominantly focusing on the occurrence and characterization of AMR. In these studies, the mere occurrence of AMR of human health concern is often interpreted as posing human exposure and health risks. However, estimating human exposure and health risks is a non-trial task for several reasons. First, there is a need for the accurate quantitative determination of the likelihood or probability of the transfer of antimicrobial-resistant organisms or resistance genes, and mobilome from the rodents, insects, and pet reservoirs and vectors to intermediate vectors or humans. Such transfer might occur via HGT facilitated by MGEs or via other human intake pathways, such as dermal contact, ingestion, and inhalation. Second, there is a need to estimate the frequency and magnitude of human exposure to AMR in such vectors. It is important to note that not all antimicrobial-resistant microorganisms and their resistance genes constitute human pathogens. The antecedent human health conditions (e.g., underlying health conditions) and age might affect human susceptibility, further complicating the estimation of human health risks.

### 4.3. Drought on Fertile Grounds? AMR in Insects, Rodents, and Pets in Developing Countries

Insects, rodents, and to some extent, pets are an integral component of societies in the developing regions of Asia, South America, and Africa [7,235]. For example, both edible insects, rodents, and other wild animals contribute significantly to household food security and nutrition [236,237]. Companion animals, including cats and dogs, are also common, where the latter is even used for household security and hunting edible wild animals. Despite the prominence of these animals, AMR in insects, rodents, and pets attracted limited research and public attention in developing countries. Yet, several risk factors might also promote the proliferation and dissemination of AMR among the environment, animals, and humans in developing countries. These risk factors include (1) a putatively high number of stray animals, including dogs and cats [238]; (2) a high burden of both animals and human diseases due to a predominantly tropical environment [10,13], resulting in a corresponding high use, misuse and abuse of antimicrobials in animal and human health [239]; and (3) terrestrial, aquatic and atmospheric systems highly polluted by chemical agents including antimicrobials, pharmaceuticals, and metals that select for antimicrobial resistance [240]. Moreover, developing countries have weak and poorly enforced animal, human, and environmental health regulations, while early detection and mitigation of AMR is challenging due to a lack of systematic surveillance systems [13]. Therefore, the lack of research in socio-economic settings with high-risk factors where conditions are ideal to understand AMR in insects, rodents, and companion animals and its health risks, is ironic. As Loewenson [241] aptly summed up, this constitutes a ‘*drought on fertile grounds*’.

### 4.4. Towards a Quantitative Human Health Risk Assessment

Quantitative microbial risk assessment (QMRA) provides an ideal tool to overcome the methodological limitations of current studies, while simultaneously enabling a quantitative determination of human exposure and health risks of AMR in rodents, insects, and pets. QMRA is a systematic approach for identifying, evaluating, and estimating the human exposure routes and potential health risks of AMR, based on the source–exposure pathway–receptor–impact framework [242]. QMRA uses quantitative dose-response relationships or models to estimate human exposure pathways and the associated health risks. In the case of AMR in rodents, insects, and pets, QMRA requires data on the following—(1) the occurrence, nature, and concentration of various AMR in rodents, insects, and pets, and their associated environmental media; (2) the transfer of AMR from sources such as rodents, insects, pets, and their associated environmental media to the receptor (humans) via multiple exposure routes (contact, ingestion, inhalation); (3) the behavior, and fate of AMR from source to the receptor, including any degradation and potential mutations; and (4) frequency and magnitude of human exposure via the various routes, and the characteristics of the human receptor, including underlying conditions, age, and antecedent health stressors. Yet currently, barring data on occurrence, the bulk of the data required for QMRA is still lacking. Thus, estimating human exposure and health risks associated with AMR in rodents, insects, and pets, and the development and validation of predictive methods for risk estimation is a key area of further research. Information from such research is critical in the development of strategies for the mitigation of potential human health risks of AMR.

## 5. Human Health Risk Assessment and Mitigation

### 5.1. Health Risk Assessment

Considering the risks arising from the spread of AMR, as outlined in the previous sections, there is a need to continuously assess the associated human health risks. The guidelines and recommendations related to the general assessment of AMR risks are outlined elsewhere [243,244]. Thus, this part focuses specifically on AMR in insects, rodents, and pets present in the household settings, specifically risk identification, and outlining management goals [245].

The risk assessment of human health threats should predominantly be based on data on the following:monitoring the usage of antimicrobial in companion animals,the extent to which the AMR occurs in pets, insects, and rodents entering households unintentionally,the association between AMR in rodents, insects, and animals of interest, and humans coming in contact with such animals, andthe routes via which AMR microorganisms and/or ARGs can be transmitted between household animals and humans.

Despite the knowledge on the role of livestock animals in the spread of AMR, companion animals, such as cats and dogs, were for long not regarded as a significant transmission of ARGs or antimicrobial-resistant microorganisms to humans. Therefore, data on human health risks traceable to AMR in insects, rodents, and pets are scarce. Hence, further research efforts are required to generate information important for human health risk assessment in this context. The limited understanding of the occurrence of AMR in companion animals and other animals kept as pets in household settings (including rodents), coupled with the poorly understood risk factors, and transmission routes make it difficult to conduct comprehensive quantitative risk assessments. This position was also highlighted by the European Medicines Agency [246]. Similarly, data to perform risk assessments related to rodents and insects are also limited and requires further studies. To accelerate the acquisition of data, the approach should not only be based on purely scientific studies but must also include regularly conducted national surveillance using standardized methodologies. However, in the case of developing countries, such national surveillances are still lacking. One should, however, note that investigations of AMR in companion animals and their owners in developing countries would not be possible, without support from not only regulatory agencies and governments but also external financial assistance.

The assessment of antimicrobial usage and AMR in cats and dogs was recently performed and reported for Belgium, Italy, and The Netherlands [24]. Despite the lower use of these pharmaceuticals compared to livestock production systems, it was concluded that companion animals might represent a source of transmission of resistance genes or resistant microorganisms to humans. Notably, the study revealed that among applied antibiotics, cats and dogs frequently received CIAs. This highlights that the assessment of human health risks must consider not only the quantity and number of pharmaceuticals in use but also the qualitative factors (e.g., type of antimicrobials). Therefore, it is pivotal to understand to what extent companion animals are receiving CIA as listed by the World Health Organization [247], and the occurrence of resistance to such CIAs. This requires systematic monitoring and reporting in different geographical areas to understand the actual proportion of microorganisms that are resistant or acquire resistance to the antimicrobial drugs of interest.

Considering the possible transmission of AMR from companion animals to humans, as reported in China in relation to colistin [248], it is important to understand the frequency of AMR in pet owners relative to non-pet owners. Such assessment should also include the collection of data on the history of antimicrobial use in both animals and humans, to evaluate the potential correlations of AMR in the two compartments. A few studies already exist in this regard, for example, in Brazil [249] and Denmark [250,251]. However, one should note that correlation does not imply causation, as multiple potential routes of AMR transmission to humans make it difficult to determine the role of insects, rodents, and pets. This is because, besides the history of antimicrobial use by humans, human exposure to AMR also occurs via food (meat, eggs, milk, vegetables, and grains), drinking water, air, and dust [243]. Therefore, more definitive conclusions on the role of rodents, insects, and pets could only be drawn from nationwide surveillance, which is still lacking in most countries. Although AMR increases healthcare costs, morbidity, and mortality in developed and developing countries, its consequences for human health are of higher threat in the latter ones. This is due to the following reasons—(1) lower quality in prescription practices, (2) inadequate education of patients, (3) limited access to diagnostics, (4) higher rate of unauthorized sale of antimicrobial, and (5) generally less control over the usage of these pharmaceuticals in humans and animals [252,253,254].

Studies on quantitative health risk assessment should also cover the identification of routes of AMR transmission from pets to owners, as this is pivotal to develop management strategies and guidelines. This is, however, a particularly challenging task that should be coupled with studies addressing the frequency of AMR in animals and humans. It is also further complicated by the fact that there can be a bilateral transmission of AMR between animal and owner, making the assessment of the initial source of these transmissions even more challenging. One approach is to gather detailed information on the history of antimicrobial use in pet owners and pets (type, dosage, time of use), level of household hygiene, type, and frequency of intimate contacts between owner and pet, and the lifestyle of pets (e.g., in case of cats whether it is free-ranging, or household-based and outgoing or not). One should also note that transmission events might not only occur during the use of antimicrobial drugs in animals, but also after the microorganisms with AMR colonize the pet, following their excretion and contamination of households [220].

Since it is virtually impossible to conduct experimental studies directly addressing the transmission of AMR from actual pets to their owners, some more definitive conclusions could be drawn from prospective studies. Such research should assess the baseline level of AMR in owner and companion animals and follow it consecutively, along with tracking the history of antimicrobial use in both compartments. To date, no such investigations were conducted that highlight the urgent need for such studies. The application of a similar assessment of human health risks related to rodents and insects that are not intentionally kept in household settings but occur temporarily, is more challenging. This is due to difficulties in collecting the animals for material sampling, although one should note that such investigations were conducted on small mammals (e.g., mice, shrews) outside household settings [36].

The results indicate that rodents and insects can be sentinels or bioindicators for AMR transmission in the environment [36]. Other studies addressing the epidemiological risks associated with rodents entering household settings were also conducted [255,256,257]. This is highly important as rodents and insects can not only acquire AMR within the household setting but also outside from different environmental sources. These sources might not be directly related to antibiotic usage but might also be implicated in the transmission to companion animals, particularly outgoing cats, due to predation [36,159]. The potential routes of transmission of AMR to humans from insects and rodents and the assessment of frequency in which they occur, can be partially addressed via the experimental approach and use of laboratory animals. Such research should evaluate how effectively the AMR microorganisms and genes can be transmitted during direct contact between insects/rodents and laboratory animals, bites, contact with excrements, dead bodies, and through consumption of edible insects, rodents, and associated human food products.

One should note that AMR is, as stated by the WHO [258], “a complex problem driven by many interconnected factors, hence single, isolated interventions have little impact, and coordinated actions are required”. This consideration is also consistent with the ‘One Health’ approach. Therefore, the assessment of health risks related to household insects, rodents, and pets, and pursuing the associated management goals can only be perceived as an accompanying yet important element of the strategy of addressing AMR issues, risk assessment, and mitigation activities. At this point, such risk assessments require more data on the frequency of AMR microorganisms in these animals, the establishment of routes of transmission, and assessment of relation with AMR observed in humans.

### 5.2. Mitigation

Mitigation measures include (1) national and international guidance and monitoring on the use of antibiotics to limit the use of CIAs and non-essential use; (2) non-clinical interventions for animal health such as good feed, pet hygiene, regular check-ups, vaccination, and control of pests to improve animal health while reducing the use of antibiotics; (3) training of veterinarians on the selection of appropriate therapy to reduce antimicrobial resistance; (4) education of owners about good hygiene practices (e.g., not letting a pet to lick owner’s face, regular hands washing); (5) raising awareness among owners to strictly adhere to recommendations on the use of antimicrobial use (timing, dosage, etc.); and (6) antimicrobial susceptibility testing for empirical guidance on the choice of antibiotics. For example, soft diets in dogs and cats are associated with high incidences of dental diseases, and harder, chewy foods requiring vigorous prehension and mastication improved dental health [211].

Different strategies of mitigation of health risks related to AMR should be developed for animals kept as pets and animals such as rodents and insects that enter household settings temporarily. Nevertheless, the general approach should include obligatory control of antimicrobial use and their release into the environment from all sources. In the case of pets, represented predominantly but not exclusively by cats and dogs, the use of antimicrobials should rely on guidelines issued to veterinary specialists and recommendations issued to pet owners via veterinary clinics. Given the fact that there are limited antimicrobials available for use in companion animals, the mitigation strategy should be of benefit to both animals and humans. Specifically, such a strategy should include:limiting the use of antimicrobials in pets only to the clinically justified applicationsraising the awareness of the complex AMR issue within the veterinary specialists and even among the pet ownerslimiting direct contact between pet owners and animals during the use of antimicrobialsusing hand hygiene practices after direct contact with petsavoiding intimate contacts with pets through face licking and sharing a bedregular cleaning of households; andwearing rubber, latex, or vinyl gloves when cleaning urine and droppings from insects, rodents, and pets

In cases where rodents are not kept intentionally in household settings (predominantly mice and rats), the following mitigation approach is proposed:blocking all potential entry routes (foundation cracks, unsealed windows, doors, etc.), particularly in the colder season when the risk of entering is the highestsealing garbage bins and containerssealing food, including pet foods, to avoid cross-contaminationcleaning the household to remove uneaten parts of foodusing traps and baits in case of a high risk of infestation; andpreventing household cats from going outside to limit the predation, potential rodent-cat transmission, and further cat-human transmission.

The mitigation strategy for the health risks related to AMR transmission via insects entering the household should include:collection of dog’s and cat’s droppings and their appropriate disposal to avoid contact with insects (e.g., flies) that feed or develop in excrements, acquisition of AMR and its further spread to humans [95].regular cleaning of householdsThe use of preventive measures to limit insects from entering the household setting (e.g., screens on windows, mosquito nets); andelimination of insects in household setting (e.g., mechanically or chemically in case of infestations)

Given that rodents, insects, and pets might co-exist, a combination of these measures might be required to mitigate the health risks associated with AMR.

## 6. Future Perspectives

### 6.1. Future Research Directions

Further research is needed to address several knowledge gaps in the context of the source-pathway-receptor-impact framework.

#### 6.1.1. Comprehensive Database on AMR in Insects, Rodents, and Companion Animals

Several insects, rodents, pets, and human ectoparasites such as lice and fleas remain understudied [259]. Hence, there is a need for further studies, including a broad range of common household and edible insects (ants, mosquitoes, lice, fleas, termites, moths, crickets), and rodents and pets (e.g., birds, reptiles, rabbits, guinea pigs). Moreover, there is a need to extend research beyond antibiotic resistance to include resistance to antimalarials, antivirals, antihelminthics, antifungals, antiprotozoals, and antiprions [1,260]. Drug-resistance to antiprions or therapeutic drugs used to treat neurodegenerative diseases caused by the accumulation of prions or self-propagating misfolded proteins were also reported [1]. However, the mechanisms of some of these forms of AMR might differ from those of antibiotic resistance.

#### 6.1.2. Partitioning AMR between Natural and Anthropogenic Pools

The relative contribution of the intrinsic and induced AMR to the AMR burden in insects, rodents, and pets is unclear. This is because partitioning AMR between the two sources is difficult due to the complex behavior of AMR and methodological limitations. Hence, research is required to develop and validate tools to partition AMR between the two sources.

#### 6.1.3. Transfer Mechanisms and Behavior in the Environment–Animal–Human Interface

The transfer of AMR in insects, rodents, and pets to humans might occur via multiple pathways including contaminated food [261,262], but the exact animal–human transmission mechanisms and behavior of AMR in each compartment remain unclear. Further work is needed to understand the key environmental and microbial drivers of the occurrence, proliferation, and transmission of AMR in the environment–animal–human interface. Such evidence is critical in the development of mitigation measures to limit the circulation or flow of AMR among the environment, animals, and humans.

#### 6.1.4. AMR Receptors and Primers in Insects, Rodents, and Companion Animals

Cell membranes and proteins of some organisms might act as receptors or primers that suppress or amplify AMR [263,264]. However, the existence and nature of such primers and receptors remain poorly understood in the case of AMR in insects, rodents, and pets. This requires further research to understand whether the differential acquisition of AMR in various insect, rodents, and companion animal species could be attributed to such receptors and primers. Bacteria communication or ‘quorum sensing’ is also critical in the exchange of AMR among bacteria species and other organisms [265], but these processes are still poorly studied in insects, rodents, and pets.

#### 6.1.5. Insects as Potential Sources of Novel Antimicrobials to Mitigate AMR

Evidence shows that insects are a potential source of insect defensins or antimicrobial peptides that provide the first line of defense against antimicrobials [266,267]. The capacity of insect antimicrobial peptides as antiviral, antibacterial, and antifungal factors is reported in the literature [268,269]. Synthetic antimicrobial peptides or polymers are attracting significant research attention as an alternative therapy to AMR compared to conventional antimicrobials [270]. Thus, scope exists to develop novel antimicrobials from such insect antimicrobial peptides, but this aspect warrants further investigation.

#### 6.1.6. Human Exposure and Health Risks via the Consumption of Edible Insects and Rodents

Despite harboring AMR, edible insects, rodents, birds, and other wild animals are sources of food and nutrition in several cultures in South America, Asia, African, and even some developed countries [7,135,271] while some insects (e.g., the Black soldier fly) are harnessed to produce novel foods [272,273]. Several studies even show that processed ready-to-eat insect food products derived from grasshoppers, crickets, and mealy worms harbor AMR of human health concern [134,135,262]. However, the contribution of AMR derived from direct consumption of edible rodents and insects and the associated food products is currently under-studied. Further studies focusing on regions with high consumption of such foods are required to investigate the potential human health risks associated with such practices.

#### 6.1.7. Quantitative Microbial Risk Assessment

The evidence linking AMR in insects, rodents, and pets remain largely inferential and qualitative. Further studies are required based on QMRA to determine the link between AMR and human health risks directly. Unlike qualitative or inferential evidence, QMRA uses quantitative dose-response models or relationships to estimate the probability of the occurrence of an endpoint [274]. In the current case, QMRA would estimate the probability of the occurrence of adverse human health outcomes as a result of exposure to AMR in insects, rodents, and pets. Such QMRA should also account for the following—(1) the nature and concentrations of the antimicrobial-resistant microorganisms and ARGs, (2) behavior and fate of the AMR, and (3) multiple transmission pathways from source to receptor (i.e., humans). Such research would also provide critical epidemiological evidence linking AMR in insects, rodents, and pets to specific human health outcomes.

#### 6.1.8. The Contribution of Insects, Rodents, and Pets to the Global Human AMR Burden

Currently, no estimates exist on the contribution of AMR in insects, rodents, and pets to the global AMR burden and the associated health risks. As Dunachie et al. [3] pointed out, estimating the global AMR burden in terms of morbidity and mortality is a non-trivial task due to methodological limitations. Therefore, the development and validation of frameworks and metrics for estimating the global contribution of AMR in various environmental compartments or resistomes, including insects, rodents, and pets, require further investigation.

#### 6.1.9. Understanding the ‘Human Factor’ in AMR

Public knowledge, perceptions, attitudes, and practices as well as those of human and animal health practitioners (i.e., the human factor) is a key driver of AMR through overuse, misuse, and abuse of antimicrobials. However, little is known about the contribution of the human factor to the transmission of AMR in the environment–animal–human interface.

#### 6.1.10. Increasing the Global Footprint of Developing Regions in AMR Research

Developing regions, including Africa, are poorly represented in the literature on AMR, particularly with respect to insects, rodents, and pets. Therefore, further research is needed to determine the nature of AMR in various species of insect, rodent, and companion pets in developing countries. Moreover, such research should address several generic knowledge gaps highlighted here.

### 6.2. Harnessing Emerging and Novel Tools to Unravel the Complex Behavior of AMR

The bulk of studies are based on conventional culturing and molecular techniques. Several recent advances in analytical tools such as computational or in-silico techniques, genomics, network analysis, and big data analytics only received a cursory application in research on AMR in insects, rodents, and pets. Yet, these emerging tools have the potential to complement existing conventional methods and provide further insights into the complex behavior of AMR. The generic applications of emerging and novel analytical tools in AMR research are discussed in an earlier paper focusing on AMR in the funeral industry [13]. Here, potential applications of the emerging and novel tools in the context of AMR in the environment–animal–human interface are highlighted.

#### 6.2.1. Genomic Tools

Genomics is a collective term referring to a range of advanced molecular techniques, including, among others, meta-transcriptomics, (exo)metabolomics, metagenomics, and proteomics [275]. Compared to cultural methods, genomics has a number of advantages relevant to investigating AMR in insects, rodents, and pets and their health risks. First, genomics can be used to study the occurrence and proliferation of AMR in non-culturable organisms in insects, rodents, pets, and humans. Moreover, genomic tools have the capacity to reveal complex human and ecological health effects, including effects of AMR on metabolic networks, gene expression profiling, protein and enzyme synthesis and functions, and trophic interactions [275,276].

#### 6.2.2. Computational or In-Silico Techniques

In-silico or computational techniques are collection modeling or simulation tools that allow the rapid computer-aided design of virtual experiments. In the context of AMR in the environmental–animal–human continuum, in-silico techniques have the following potential applications—(1) rapid prototyping and analysis of complex systems including metabolic and AMR networks, which are difficult to study via conventional research methods, and (2) outputs of in-silico analysis can be used to inform the cost-effective design of laboratory and field experiments, and subsequent analysis of data.

#### 6.2.3. Network Analysis

The occurrence, dissemination, and fate of AMR in the various compartments form a complex network that is often difficult to disentangle using conventional tools. One question that arises is, ‘*Does AMR in humans, insects, rodents, and pets originate from the same environmental pool or resistome?*’. Network analysis is a powerful tool used to understand complex interactions among various components. Thus, a combination of network analysis and genomics can be used to unravel such complex interactions and reveal the degrees of similarities of AMR in the various compartments or resistomes.

#### 6.2.4. Big Data Analytics

A comprehensive understanding of the occurrence and behavior of AMR requires analysis of various media. These include (i) environmental media (soil, wastewaters, drinking water, and ambient air), (ii) animals including insects, rodents, pets, and their prey and predators, and (iii) human media such as feces, gut specimen, and serum. Such efforts will generate large quantities of data (i.e., ‘big data’) that are often difficult to analyze and visualize using conventional statistical tools. Big data analytics such as machine learning might provide ideal tools for the extraction of subtle trends and patterns in such large datasets encompassing AMR in environmental, animal, and human media and their potential human and ecological health risks.

The application of these emerging tools could significantly improve our understanding of the complex behavior and health risks of AMR. Research applying a combination of these emerging tools represents the next frontier in AMR research in the environment–animal–human continuum.

## 7. Conclusions and Outlook

The present review applied the source-pathway-receptor-impact framework to track the occurrence and circulation of AMR in household and edible insects, rodents, and pets. Insects and rodents acquire antimicrobial resistance via foraging of prey harboring AMR, which in turn proliferate and persist in various compartments of insects, rodents, and pets, including the skin and gut systems. The widespread use of antimicrobial treatments in veterinary medicines and pet foods also induce AMR micro-organisms found in pets. Thus, several household insects (e.g., houseflies, cockroaches) and edible ones (e.g., crickets, grasshoppers), rodents (rats, mice), and pets (dogs, cats), harbor and act as reservoirs of antimicrobial resistance. To date, antimicrobial-resistant microorganisms and their corresponding ARGs for a diverse range of first-line and last-resort antimicrobials, including antibiotics, are reported in insects, rodents, and pets. Subsequently, household insects, rodents, and pests act as vectors that disseminate antimicrobial resistance into other environmental compartments, including humans via direct contact, as prey for other organisms, human food contamination, and horizontal gene transfer. Thus, household insects, rodents, and pets might act as sentinels or bioindicators for the surveillance of AMR. In this regard, insects, rodents, and pets could be subsequently used as an early warning system for the surveillance of AMR ahead of its detection in humans.

Human exposure and health risks, and key risk factors were discussed, including those unique to low-income countries. Current evidence on human exposure and health risks is largely inferential and qualitative. Hence, comprehensive data based on quantitative microbial risk assessment are still lacking, making it difficult to trace specific human health risks or outcomes to AMR in insects, rodents, and pets. To safeguard human health, mitigation measures are proposed based on the one health approach. Future research, including ten knowledge gaps, were highlighted, including (1) quantitative microbial risk assessment, (2) investigating AMR in common household insects, rodents, and pets in developing countries, (3) partitioning antimicrobial resistance in insects, rodents, and pest between anthropogenic and natural sources, and (4) use of emerging and novel techniques (e.g., in-silico methods, genomics, network analysis, and big data analytics or machine learning) to understand the role of household insects, rodents, and pets in the persistence, circulation, and health risks of AMR. These emerging tools can complement conventional ones to address the highlighted knowledge gaps and improve our understanding of the occurrence and circulation of AMR in the environment–animal–human continuum.

## Figures and Tables

**Figure 1 antibiotics-10-00068-f001:**
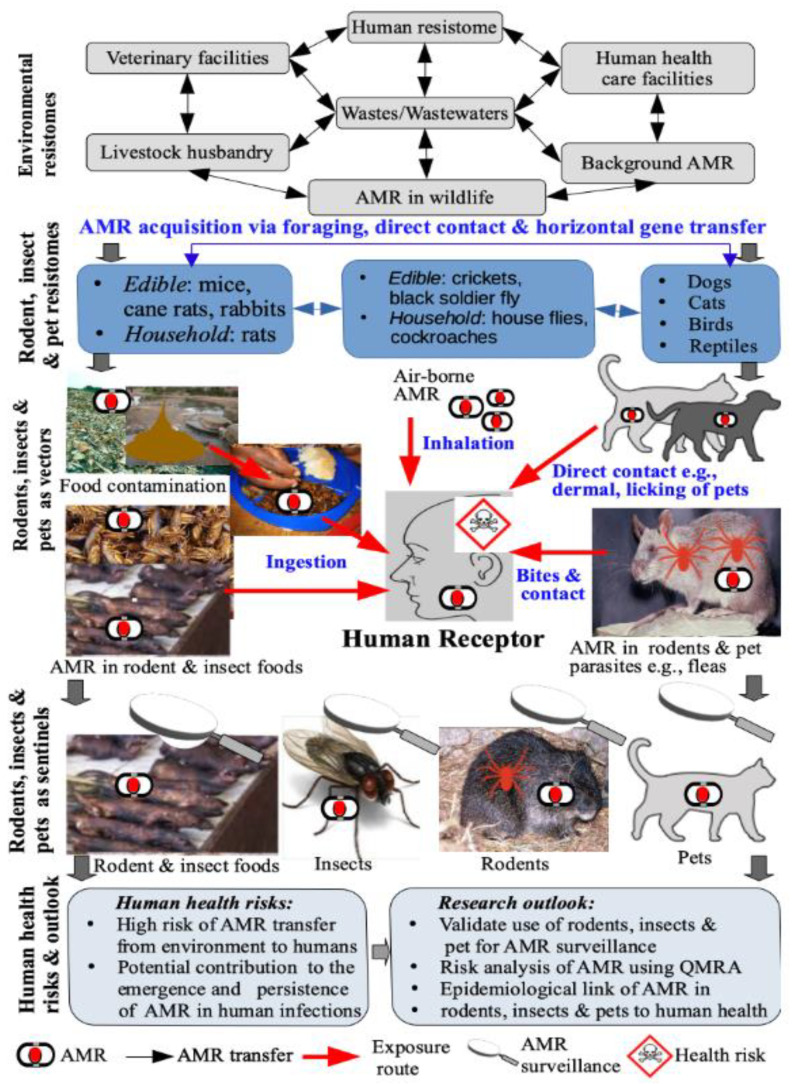
Source–pathway–receptor depiction of the occurrence, circulation, and human health risks of antimicrobial resistance (AMR) in rodents, insects, and pets. The arrows depict the complex exchange of AMR among the various reservoirs or resistomes.

**Figure 2 antibiotics-10-00068-f002:**
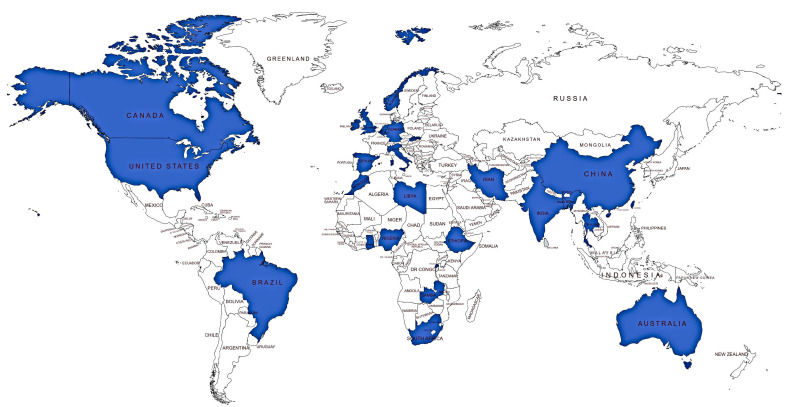
Countries (shaded) that reported AMR in insects between 2010–2020. The search was based on PubMed Central and Google Scholar databases.

**Figure 3 antibiotics-10-00068-f003:**
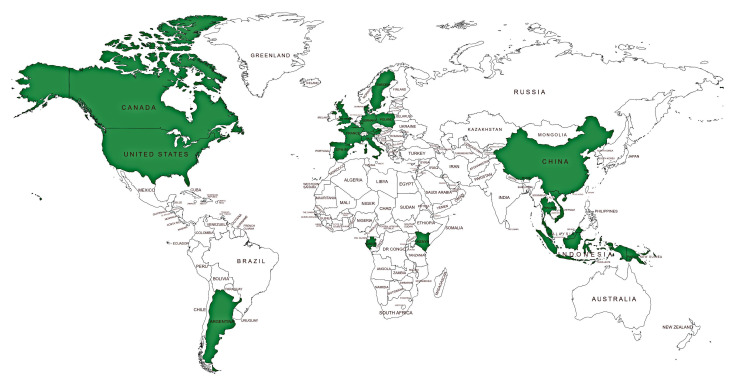
Countries (shaded) that reported AMR in rodents in the period 2010–2020. The search was based on PubMed Central and Google Scholar databases.

**Figure 4 antibiotics-10-00068-f004:**
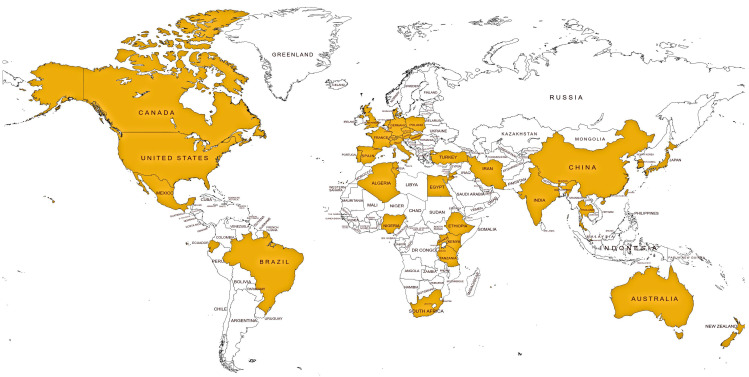
Countries (shaded) that reported AMR in pets in the period 2010–2020. The search was based on PubMed Central and Google Scholar databases.

**Table 1 antibiotics-10-00068-t001:** A summary of antimicrobial resistance reported in some insects.

Insect Name	AMR Organism	AMR Type	ARG Detected	Reference
Houseflies (*Musca domestica*) Cockroaches (*Blattaria/Blattodea*)	Enterococci (*E. casseliflavus; E. hirae; E. faecium; E. faecalis*)	Tetracycline, Erythromycin	Tetracycline tet(M) Erythromycin erm(B) Tn*916/1545* transposon family, gelatinase gelE, esp, and asa1	[37]
			tet(M) and erm(B) and Tn*916/1545* transposon family	
Cockroaches (*Periplaneta Americana* *Blattella germanica*)	*38 species of gram-negative bacteria, 20 species of glucose non-fermenter bacilli and 6 species of gram-positive bacteria*	Ampicillin; Gentamicin; Ciprofloxacin, Ofloxacin; Chloramphenicol Tetracycline; Trimethoprim-sulfamethoxazole; Penicillin; Streptomycin; Erythromycin; Oxacillin; Vancomycin; Cephalothin; Ceftazidime; Imipenem; Piperacillin; Cefoperazone	Not determined	[43]
Houseflies (*Musca domestica*)	*E. faecalis*	Tetracycline; Erythromycin; Streptomycin; Ciprofloxacin; Kanamycin	transposon Tn*916* and members of the Tn*916*/Tn*1545* family	[36]
Houseflies (*Musca domestica*)	*E. coli*	Tetracycline; Ampicillin; Streptomycin; Sulfonamides; Trimethoprim-Sulfamethoxazole; Chloramphenicol; Nalidixic acid	Tetracycline tetA and tetB, sulphonamide sul1, sul2, sul3, extended-spectrum b-lactamase bla_TEM_, strA	[142]
Stored-product grain insects, e.g., darkling beetle, A. diaperinus; Lesser grain borer, *Rhyzopertha dominica*; Foreign grain beetle, *Ahasverus advena*; red flour beetle, *Tribolium castaneum (Herbst*); warehouse beetle, *Trogoderma variabile Ballion*;	*E. casseliflavus*; *E. gallinarum*; *E. faecium*; *E. faecalis*; *E. hirae*	Tetracycline; Streptomycin; Erythromycin; Kanamycin; Ciprofloxacin; Ampicillin; Chloramphenicol	gelatinase gelE; enterococcal surface protein esp; cytolysin cylA	[143,144]
Ants (*Tapinoma melanocephalum* (*Fabricius*) and *Camponotus vittatus* (*Forel*)	Coagulase-positive *Staphylococcus (S. aureus*); Coagulase-negative *Staphylococcus*; Gram negative Bacilli	Cephalotine; Oxacillin; Penicillin; Tetracycline; Vancomycin; Ampycillin; Cephalotine; Ciprofloxacine; Sulphazotrin	Not determined	[145]
Gypsy moth larvae (*Lymantria dispar* L.)	*Enterococcus* spp.; members of the *Enterobacteriaceae*	Carbenicillin; Ceftazidime; Gentamicin; Erythromycin; Kanamycin; Streptomycin; Vancomycin; Chloramphenicol; Rifampin; Nalidixic acid; Tetracycline.	ramA; sdeX; sdeY; bla_LRG-1_	[146]
Bedbugs (*Cimex lectularius* L.)	*E. faecium*, *S. aureus*	Vancomycin; Methicillin; Ampicillin; Teicoplanin; Aminoglycosides Erythromycin	Not determined	[147]
Honeybees (*Apis mellifera* L.)	*Snodgrassella alvi (Betaproteobacteria) Alphaproteobacteria*	Tetracycline Oxytetracyline	tetB, tetC, tetD, tetH, tetL, tetY tetM tetW	[81]
Houseflies (*Musca domestica*) False stable flies (*Muscina stabulans*)	*E. coli*	Tetracycline; Streptomycin; Kanamycin; Ampicillin; Cefazolin; Cefpodoxime; Trimethoprim	extended-spectrum b-lactamase (ESBL) bla_CTX-M-15_	[65]
Flea midgut (*Xenopsylla cheopis*)	*Yersinia pestis*	Streptomycin; Gentamycin; Tetracycline; Chloramphenicol; Sulphonamides	Not determined	[111]
Houseflies (*Musca domestica*) blow-flies (Lucilicia species) and Bottle flies (*Phaenicia* species)	Enterococci (*E. casseliflavus; E. gallinarum; E. faecium; E. faecalis) and* Staphylococci (*S. saprophyticus; S. aureus;* *S. xylosus; S. epidermidis*)	Penicillin, Quinupristin-dalfopristin; Erythromycin; Tetracycline; Clindamycin	erm(B); erm(A); msr(C); msr(A/B); transposon Tn*916*	[66]

**Table 2 antibiotics-10-00068-t002:** Percentage resistance to antimicrobial agents by different bacterial species isolated from rodents.

Rodent Species	Microbial Species	AMR Profile	Comments	Reference
Rat species 1	*S. aureus*, *E. coli* *Pasteurella pnemutropica*	Amp (75), Pen (75), AM/Cl (75), Te (12.5) Amp (11.1), Pen (100), Te (25) Amp (11.1), Pen (33.3), AM/Cl (22.2), Te (11.1)	Number in brackets equals % resistance	[176]
Mice species 1	*S. aureus*, *E. coli* *P. pnemutropica*	Amp (87.5), Pen (8.5), AM/Cl (37.5), Te (12.5) Pen (8.5) Pen (20)	Number in brackets equals % resistance	[176]
Rat species 2	*E. coli*	Amp (23.3), Strep (15), Tmc (6.6), Te (3.3), Amc (1.7)	Number in brackets equals % resistance	[177]
*Apodemus* *sylvaticus*	*E. coli*	Amp (48) (21) Tm (33) (8) Cf (21) (4) Ctx (18) (0)	No//in brackets indicate no//of animals with resistant *E. coli* over no//of animals trapped in coastal vs. inland habitats	[36]
Rat species 3	*E. coli* *K. pneumonia* *Pseudomonus paucimobilis* *Chryseomonas luteola* *Aeromonas caviae* *Burkhoddria cepacia*	Te (16.7), Gn (16.7), Apr (16.7), S3 (66.6), C (83.3), Crd (66.6), Cxm (33.3), Amp (50), Na (50), Tm (50) S3 (50), C (100), Crd (100), Amc (100), Cxm (50), Amp (100), Tm (100) Te (12.9), S3 (9.7), C (12.9), Ctx (67.7), Crd (3.2), Amc (9.7), Cxm (16.1), Amp (6.5), Na (41.9), Tm (6.5) Te (15.4), Gn (7.7), Apr (7.7), S3 (23.1), Ctx (92.3), Crd (53.8), Amc (46.2), Cxm (53.8), Amp (4.2), Na (77), Tm (46.2) Te (37.5), Gn (25), S3 (50), Ctx (50), Crd (25), Amc (37.5), Cxm (62.5), Amp (62.5), Na (87.5), Tm (87.5) Te (66.6), Gn (66.6), S3 (66.6), Ctx (66.6), Amc (16.7), Cxm (83.3), Amp (66.6), Na (100), Tm (66.6)	Number in brackets equals % resistance	[178]
Wild rodents	*Hafnia alvei* *E. coli* *Serratia liquefaciens*	Te (76), Tm (10), Na (24), Ac (98), Ap (95), Cxm (100) Te (14), Na (9), Ac (97), Ap (89), Cxm (100) Te (63), Tm (30), Na (30), Ac (100), Ap (97), Cxm (90)	Number in brackets equals % resistance	[159]
Wild rodents	*Alcaligenes* spp. *Serratia fonticola* *Enterobacter intermedius* *Enterobacter amnigenus* *Cedacae davisiae* *Providencia rustigianii*	Te (44), Tm (67), Na (56), Ac (67), Ap (67), Cxm (78) Te (50), Na (22), Ac (72), Ap (94), Cxm (67) Te (39), Tm (23), Na (23) Ac (85), Ap (92), Cxm (77) Te (50), Na (40), Ac (90), Ap (100), Cxm (90) Te (44), Tm (22), Na (22), Ac (67), Ap (89), Cxm (89) Te (17), Tm (17), Na (17), Ac (100), Ap (83), Cxm (67)	Number in brackets equals % resistance	[159]
*Rattus rattus*, *Rattus norvegicus*, *Mus musculus*	*E. coli*	Amp (35), Te (15), Ap (10), Na (10), C (5), S3 (5), Cf (5), Nf (5)	Number in brackets equals % of isolates showing resistant phenotype	[161]

Antibiotics: Amp, ampicillin; Pen, penicillin; AM/Cl, amoxicillin with clavulanic acid; Te, tetracycline; Strep, streptomycin; Tmc, Co-trimoxazole; Amc, Co-amoxiclav; Tm, trimethoprim; Cf, Ciprofloxacin; Ctx, Cefotaxime; Gn, Gentamicin; Apr, Apramycin; S3, sulfamethoxazole; C, Chloramphenicol; Crd, cephradine; Cxm, cefuroxime; Na, nalidixic acid; Ap, amoxicillin; Ac, amoxicillin/clavulanic acid; Cm, chloramphenicol; and Nf, Norfloxacin.

**Table 3 antibiotics-10-00068-t003:** Some common bacterial diseases of companion animals, including those resistant to antimicrobials.

Pet	Disease	Causative Bacteria	Reference
Dogs	Urinary tract infections,	*E. coli, Staphylococcus* spp. (*S. intermedius, S. aureus), P. aeruginosa, E. faecalis, Proteus mirabilis*	[202]
	Dental	*Actinom-vces* spp., *Streptococcus* spp. and other species	[203]
	Canine infectious respiratory disease (Kennel Cough)	*several viruses/bacteria* *Streptococcus equi subsp. zooepidemicus* *Bordetella bronchiseptica* *Mycoplasma ureaplasma*, *Mycoplasma acholeplasma*, *Mycoplasma cynos*	[204,205,206]
	Pasteurellosis	*Pasteurella* spp.	
	Bacterial pneumonia	*Enterobacteriaceae* (e.g., *E. coli*, *Enterobacter* spp., *Klebsiella* spp.), *Pasteurella* spp., *Bordetella bronchiseptica, Streptococcus* spp. *Clostridium tetani*	[204,207]
	Tetanus (cats)		[208]
Cats Horses	Urinary tract infections Chlamydiosis Tetanus (cats) Strangles Botulism Tetanus (Lockjaw)	*Chlamedia felis* *C. tetani* *Streptococcus equi* *Clostridium botulinum* *C. tetani*	[208,209]

**Table 4 antibiotics-10-00068-t004:** Some examples of antimicrobial-resistant bacterial isolates from pets in different countries. The percentage in brackets represents the prevalence of AMR isolates where data are available.

Pet	Resistance	Bacteria	Country	Reference
Dogs, Cats	Ampicillin (18%)	*E. coli*	Belgium, Italy, Netherlands	[24]
Dogs	Imipenem (0.2%), colistin (5%)	*K. pneumoniae*	Portugal	[218]
	Methicillin	*S. aureus*		[209]
Dogs, cats	Cefotaxime (66–75%) Ceftazidime (71–80%)	*E. coli*	Italy	[197]
Dogs	Cefazolin (43%), fluoroquinolone (22%)	*E. coli*	Taiwan	[219]
Dogs, Cats	chloramphenicol, tetracycline, doxycycline, co-trimoxazole, ampicillin, cefotaxime	*Klebsiella* spp.	India	[198]
	amoxicillin (4.8%), ampicillin (21.2%), lincomycin (98%), tetracycline (95%)_enrofloxacine(77%), ofloxacin (64%), ciprofloxacin (73%)	*Enterococcus* spp. (*E. faecium*, *E. avium*, *E. faecalis*)	Portugal	[196]
Dogs, Cats	Methicillin	*Staphylococcus pseudintermedius* (63%), *S. aureus* (50%)	Singapore	[220]
	Fluoroquinolone	*E. coli* (40%)		[220]
	Carbapenem	*K. pneumoniae* (7%)		[220]
Cats	Ampicillin (42%), amoxicillin/clavulanate (53%), erythromycin 40%), tetracycline (24%), ciprofloxacin (63%), teicoplanin, vancomycin (24%)	*E. faecium, E. faecalis*	Italy	[197]
Horses	Cephalosporin, ciprofloxacin Gentamycin, tobramycin	*K. pneumoniae*	Austria	[199]
Horses	Several antibiotics	*E. coli*	USA	[221]

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
