# Peer review of "Insects, Rodents, and Pets as Reservoirs, Vectors, and Sentinels of Antimicrobial Resistance"

_antibiotics, 2021, doi:10.3390/antibiotics10010068_

Round 1
Reviewer 1 Report
Review
Insects, rodents, and pets as reservoirs, vectors, and sentinels of antimicrobial resistance
Gwenzi et al.
The review covers very important and not well-studied topic, therefore it deserve to be published. Before it acceptance, the manuscript requires a revision. Main remark concerns information about the parasites, protozoa and other organisms, which do not follow directly the topic for antibacterial resistant. This information should be deleted. Other remarks are listed below.
Line 37: machine learning – what is the meaning? Please, clarify.
Keywords: Resistance to antibiotics; it is too repetitive, please delete it; antimicrobial resistance genes; companion animals; human exposure pathways; human health risks; quantitative microbial risk assessment
Lines 47-49: This sentence contains wrong information. Please, revise it. There are two options:
First, Line 48: antimicrobial is not correct, therefore delete it
Second: I would suggest deleting this sentence because the aim of the review is to deal with resistance to antimicrobials
Line 121: antimicrobial is not necessary to be written here, therefore delete it
Line 137: ARG - antimicrobial resistance genes??? – explain the abbreviation
Line 139: ABR - explain the abbreviation
Line 202: may be the authors had in mind “standard culture medium for MIC determination”, please, clarify
Lines 278-299: Already in this paragraph, the authors should acknowledged that the spread of AMR can be occurred also in opposite direction: from humans to animals; from flies/cockroaches from human hospitals to animals. Please add necessary information, the authors already wrote “one world, one health”
Table 1: Please, restructure the table by adding horizontal line because the feeling is that in some places the data are mixed and it is not easy for readers to understand the information
Lines 332-337: Suddenly the authors started to talk for parasites. The information is correct but it is not within the aim of this review. Please, delete it.
Line 340, Line 375, Line 377, Line 470, Line 641, Line 721, Line 731, Line 944 and elsewhere: antimicrobial is not necessary to be written here, therefore delete it
Line 374: “3.1.1. How insects acquire antimicrobial resistance” – written in this way the subheading is wrong. I would suggest: “3.1.1. How insects acquire bacteria that caring genes for antimicrobial resistance” or “3.1.1. How insects acquire resistant bacteria”. Additionally this paragraph suffers from lack of information about the flies/ cockroaches from human hospitals or other origin related to humans. Some information is included in the next paragraph on the same page but nevertheless few sentences can be added according to the current remark.
Table 2: the same remark as for Table 1.
The authors can add 3-4 sentences and to try to summarize the information about the resistance against the different groups of antibiotics. Again, the reader will easily take the message in this way.
Line 685: replace “microbials” with “microorganisms”
Table 4: E. coli – there is typos mistake. Please, also consider the same remark as for Tables 1 and 2
Lines 1138-1145: again only the role of the vet is mentioned, but it would be better to mention what can be improved from human side in relation to prescribing and usage of antibiotics.
Section 6. Future perspectives: The information from this section was outlined to certain extend in the previous parts of the manuscript. Therefore I would suggest presenting the information more concisely.
Author Response
Review No. 1
Insects, rodents, and pets as reservoirs, vectors, and sentinels of antimicrobial resistance
Gwenzi et al.
The review covers very important and not well-studied topic, therefore it deserve to be published. Before it acceptance, the manuscript requires a revision. Main remark concerns information about the parasites, protozoa and other organisms, which do not follow directly the topic for antibacterial resistant. This information should be deleted. Other remarks are listed below.
Authors’ Response:
-
We thank the reviewer for the overall positive remarks and the detailed comments meant to improve our work. We have considered and addressed all the comments raised as explained in detail below.
Line 37: machine learning – what is the meaning? Please, clarify.
Authors’ Response:
-
We agree and revised to ‘big data’ analytical tools which is trending and familiar to most readers. What ‘big data’ analytical tools entail and how they can be used to understand AMR is explained in detailed in the manuscript under Future perspectives. Hence, we avoid the details in the Abstract.
Keywords: Resistance to antibiotics; it is too repetitive, please delete it; antimicrobial resistance genes; companion animals; human exposure pathways; human health risks; quantitative microbial risk assessment
Authors’ Response:
-
We agree. Antibiotic resistance has been deleted
Lines 47-49: This sentence contains wrong information. Please, revise it. There are two options:
First, Line 48: antimicrobial is not correct, therefore delete it
Second: I would suggest deleting this sentence because the aim of the review is to deal with resistance to antimicrobials
Authors’ Response:
-
We considered the comment, and deleted the statement below:
-
‘ It encompasses resistance to the following groups of antimicrobial agents: (1) antibiotics, (2) antifungals, (3) antivirals, (4) anthelmintics, (5) antiprotozoals, and recently antiprions [1].’
-
We then revised ‘antimicrobials’ to ‘antimicrobial agents including antibiotics’
Line 121: antimicrobial is not necessary to be written here, therefore delete it
Authors’ Response:
-
We agree. This has been deleted
Line 137: ARG - antimicrobial resistance genes??? – explain the abbreviation
Authors’ Response:
-
We agree, and this has been defined as ‘antimicrobial resistance genes’.
Line 139: ABR - explain the abbreviation
Authors’ Response:
-
We agree, In fact, ‘ABR’ has been changed to “antimicrobial resistance”
Line 202: may be the authors had in mind “standard culture medium for MIC determination”, please, clarify
Authors’ Response:
-
We considered the comment. The intended meaning is standard protocols for antimicrobial susceptibility testing, and we revised to reflect the intended meaning:
‘..then subjected to standard protocols for antimicrobial susceptibility testing [32,33,39,40,55-57]. To identify antibiotic susceptible microorganisms or ARB in a sample, the minimum inhibitory concentrations (MIC) of antibiotics to particular bacteria are predefined [58]. ’
Lines 278-299: Already in this paragraph, the authors should acknowledged that the spread of AMR can be occurred also in opposite direction: from humans to animals; from flies/cockroaches from human hospitals to animals. Please add necessary information, the authors already wrote “one world, one health”
Authors’ Response:
-
We agree and revised as follows:
‘Conversely, AMR can be transmitted from humans to animals, and from insects such as flies and cockroaches from human hospitals to animals. This complex and continuous exchange of AMR among the various compartments or resistomes, which also applies to rodents and pets, underpins the ‘one world, one health’ concept.’
Table 1: Please, restructure the table by adding horizontal line because the feeling is that in some places the data are mixed and it is not easy for readers to understand the information
Authors’ Response:
-
We agree. Horizontal lines have been added as suggested.
Lines 332-337: Suddenly the authors started to talk for parasites. The information is correct but it is not within the aim of this review. Please, delete it.
Authors’ Response:
-
We agree and deleted the statement below:
‘External surfaces of cockroaches, houseflies, blow-flies, and bottle flies were also contaminated with human intestinal parasites in Nigeria [47] and the United States of America [66,68].’
-
The statement below was rephrase and moved to the section on rodents:
‘In this regard, there are insects such as the blood-sucking bed bugs, which have a long history of association with humans, but information on their capacity to transmit AMR and human infections is still poorly understood. However, such insects may acquire AMR and act as conduits for its subsequent transmission to humans. A typical example is the emergence of antibiotic-resistant Yestinia pestis in bubonic plague patients in Madagascar, which was linked to horizontal gene transfer from rat fleas [48].’
In section on rodents this has been revised as follows:
‘Indirectly, rodents and even pets may harbor parasites possessing antimicrobial resistance, which may be transferred to humans via such parasites. A typical example is the emergence of antibiotic-resistant Yestinia pestis in bubonic plague patients in Madagascar, which was linked to horizontal gene transfer from rat fleas [48]. The role of parasites in the transfer of AMR from rodents and pets to humans and vice versa further further demonstrates the complexity of AMR. Yet this aspect is relatively less studied compared to the direct role of rodents and pets n the persistence and transmission of AMR.’
Line 340, Line 375, Line 377, Line 470, Line 641, Line 721, Line 731, Line 944 and elsewhere: antimicrobial is not necessary to be written here, therefore delete it
Authors’ Response:
-
We agree, and these have been corrected where appropriate
Line 374: “3.1.1. How insects acquire antimicrobial resistance” – written in this way the subheading is wrong. I would suggest: “3.1.1. How insects acquire bacteria that caring genes for antimicrobial resistance” or “3.1.1. How insects acquire resistant bacteria”.
Authors’ Response:
-
We totally agree. This has been changed to “How insects acquire resistant bacteria”
Additionally this paragraph suffers from lack of information about the flies/ cockroaches from human hospitals or other origin related to humans. Some information is included in the next paragraph on the same page but nevertheless few sentences can be added according to the current remark.
Authors’ Response:
-
We agree, but this comment has been addressed in earlier responses as follows:
‘Conversely, AMR can be transmitted from humans to animals, and from insects such as flies and cockroaches from human hospitals to animals. This complex and continuous exchange of AMR among the various compartments or resistomes, which also applies to rodents and pets, underpins the ‘one world, one health’ concept.’
Table 2: the same remark as for Table 1.
Authors’ Response:
-
We agree, and this has been corrected.
The authors can add 3-4 sentences and to try to summarize the information about the resistance against the different groups of antibiotics. Again, the reader will easily take the message in this way.
Authors’ Response:
-
We agree and summarized as follows for the various animals:
Insects:
Table 1 shows that insects harbor micro-organisms possessing genes conferring resistance to a wide range of antimicrobials including ampicillin, tetracycline, streptomycin, ciprofloxacin, gentamicin, sulfamethoxazole, chloramphenicol, penicillin, and kanamycin. Resistance to last-resort antimicrobials such as cephalosporin and colistin has also been reported in Enterobacteriaceae from flies in Thailand [71].
Rodents:
In summary, microorganisms in rodents harbor genes conferring resistance to ampicillin, penicillin, amoxicillin with clavulanic acid, tetracycline, streptomycin, co-trimoxazole, co-amoxiclav, trimethoprim, ciprofloxacin, cefotaxime, gentamicin, apramycin, sulfamethoxazole, chloramphenicol, cephradine, cefuroxime, nalidixic aci, amoxicillin chloramphenicol, and orfloxacin (Table 2).
Pets
Microorganisms in pets have been reported to harbour resistance to first-line and last-resort antimicrobials including ampicillin, imipenem, colistin, methicillin, cefotaxime, ceftazidime, cefazolin, fluoroquinolone, chloramphenicol, tetracycline, doxycycline, co‐trimoxazole, cefotaxime, amoxicillin, lincomycin, enrofloxacine, ofloxacin, ciprofloxacin, carbapenem, amoxicillin/clavulanate, erythromycin, teicoplanin, vancomycin, cephalosporin, gentamycin, and tobramycin (Table 4).
Line 685: replace “microbials” with “microorganisms”
Authors’ Response:
-
We agree, and this has been replaced.
Table 4: E. coli – there is typos mistake. Please, also consider the same remark as for Tables 1 and 2
Authors’ Response:
-
We agree, and this has been corrected.
Lines 1138-1145: again only the role of the vet is mentioned, but it would be better to mention what can be improved from human side in relation to prescribing and usage of antibiotics.
Authors’ Response:
-
We agree. This section was extended to highlight the importance of good hygiene practices in owners as well as adherence to recommendations on the use of antimicrobial agents. The need for regular check-ups and vaccinations of pets was also added (See revised Section 5.2).
Section 6. Future perspectives: The information from this section was outlined to certain extend in the previous parts of the manuscript. Therefore I would suggest presenting the information more concisely.
Authors’ Response:
-
We agree. We revised and shortened by deleting the aspects already covered in earlier sections. However, the background to the knowledge gaps were only retained in cases where we feel it is critical to enhance the readers’ understanding of the motivation of the research gap (See revised Future Perspectives).
Reviewer 2 Report
This review entitled “Insects, rodents, and pets as reservoirs, vectors, and sentinels of antimicrobial resistance” revisited almost all literature regarding antimicrobials resistance involving insects, rodents and pets. The topic is very interesting. The manuscript was well organized, and written.
Minors:
- Latin names should be written in italic.
For example, Lines 417, 424, 426: E. coli and Line 522: Bacillus thuringiensis. Please also correct others in throughout the text.
- Be consistent in using Latin names. When first appeared in the text, use full names, use abbreviated genus names after. For example, Lines 458-9, Escherichia coli should be E. coli
- Line 429, “…to the eggs, larva, pupae, and finally, ..” here larva should be larvae
Author Response
Reviewer No. 2:
This review entitled “Insects, rodents, and pets as reservoirs, vectors, and sentinels of antimicrobial resistance” revisited almost all literature regarding antimicrobials resistance involving insects, rodents and pets. The topic is very interesting. The manuscript was well organized, and written.
Authors’ Response:
-
We thank the reviewer for the positive remarks and comments. We have considered and addressed the minor comments raised in the revised manuscript.
Minors:
-
Latin names should be written in italic.
For example, Lines 417, 424, 426: E. coli and Line 522: Bacillus thuringiensis. Please also correct others in throughout the text.
Authors’ Response:
-
We agree. The names have been corrected throughout the manuscript.
-
Be consistent in using Latin names. When first appeared in the text, use full names, use abbreviated genus names after. For example, Lines 458-9, Escherichia coli should be E. coli
Authors’ Response:
-
We agree. All the names have been corrected as suggested.
-
Line 429, “…to the eggs, larva, pupae, and finally, ..” here larva should be larvae
Authors’ Response:
-
We agree, and “larva” has been changed to “larvae”
Reviewer 3 Report
This is a well-written and nicely illustrated review by Gwenzi et al. on the antimicrobial resistance related to insects, rodents, and pets, with a particular emphasis on the limitation of previous studies. The topic is current due to the challenges associated with antimicrobial resistance. The review should be a useful starting point for researchers in this area. I only have the following minor suggestions:
- The acronyms should be spelled out at first mention, such as ABR and ARG.
- Page 3, Figure 1: Please provide some more interpretation in the figure legend. This figure is hard to follow by a layperson. Besides, rodents and pets also have connections in the “rodent, insect & pet resitomes” section. For example, many cats become ill from toying with a dying rat or mouse.
- Page 10, line 329: authors should change “feacal” to “fecal.”
- Page 13, lines 522 and 523: authors should correct the font size of references.
- Several scientific names of species are not italicized in the manuscript, such as line 451 and 522.
- Antarctica exists only in Figure 4 and has been excluded from Figures 2 and 3. The authors should correct it.
Author Response
Reviewer No. 3:
This is a well-written and nicely illustrated review by Gwenzi et al. on the antimicrobial resistance related to insects, rodents, and pets, with a particular emphasis on the limitation of previous studies. The topic is current due to the challenges associated with antimicrobial resistance. The review should be a useful starting point for researchers in this area.
Authors’ Response:
-
We thank the reviewer for the positive remarks and comments meant to improve our work. We have considered and addressed the comments raised n the revised version.
I only have the following minor suggestions:
-
The acronyms should be spelled out at first mention, such as ABR and ARG.
Authors’ Response:
-
We agree, and this has been corrected.
-
Page 3, Figure 1: Please provide some more interpretation in the figure legend. This figure is hard to follow by a layperson. Besides, rodents and pets also have connections in the “rodent, insect & pet resistomes” section. For example, many cats become ill from toying with a dying rat or mouse.
Authors’ Response:
-
We agree and revised as suggested. However, the aspects depicted in Figure 1 are discussed in detail in the main manuscript, and the figure also contains a key explaining what the various symbols and arrows used represent to enable the reader to understand it without the need to refer t the main text.
-
In summary, to make Figure 1 more understandable to a layperson and to address the reviewer’s comment we revised as follows:
-
Included a double arrow connecting pets and rodents (See Revised Figure 1). The connections between the other resistomes are already shown using the existing arrows. We intentionally avoided including all the possible interactions to avoid making the figure difficult to understand.
-
Revised the legend to indicate that the arrows depict the complex exchange of AMR among the various reservoirs as follows:
‘The arrows depict the complex exchange of AMR among the various reservoirs or resistomes.’
-
Revised the main text in the Introduction as follows:
‘Figure 1 presents a summary of the focus of the present paper. Specifically, the various environmental reservoirs of AMR or resistomes include solid waste and wastewater systems, animal and human health care facilities, livestock production systems, and wildlife, among others. The insect, rodent, and pet resistomes or reservoirs are also shown including the exchange of AMR among them via various processes including horizontal gene transfer. These reservoirs or resistomes act as both sources and sinks of AMR, resulting in complex circulation of AMR among the various compartments. Human exposure to AMR via ingestion of contaminated food, inhalation of contaminated particulate matter and ambient air, and direct contact and bites is also shown. Moreover, the occurrence of AMR in pets, rodents, and insects make them deal sentinels for AMR surveillance. Finally, the human health risks, and future research needs are summarized.’
-
Page 10, line 329: authors should change “feacal” to “fecal.”
Authors’ Response:
-
We agree, checked the whole document and corrected as suggested.
-
Page 13, lines 522 and 523: authors should correct the font size of references.
Authors’ Response:
-
We agree. The font sizes have been corrected throughout by the journal during formatting process.
-
Several scientific names of species are not italicized in the manuscript, such as line 451 and 522.
Authors’ Response:
-
We agree. The names have now been written in italics throughout the manuscript
-
Antarctica exists only in Figure 4 and has been excluded from Figures 2 and 3. The authors should correct it.
Authors’ Response:
-
We agree. This has been corrected in all the figures (See Revised Figures 2 and 3).